# Climate change effects on the hydrology of the headwaters of the Tagus River: implications for the management of the Tagus-Segura transfer

Francisco Pellicer-Martínez[1] and José Miguel Martínez-Paz[2]

[1]Department of Civil Engineering. Catholic University of Murcia (Spain)
[2]Department of Applied Economics. University of Murcia (Spain)

**Correspondence:** F. Pellicer-Martínez (fpellicer@ucam.edu)

**Abstract.** Currently, climate change is a major concern around the world, especially because of the uncertainty associated with its possible consequences for society. Among them, fluvial alterations can be highlighted in basins whose flows depend on groundwater discharges and snow melt. This is the case of the headwaters of the Tagus River Basin, whose water resources, besides being essential for water uses within this basin, are susceptible to being transferred to the Segura River Basin (both basins are in the Iberian Peninsula). This work studies the possible effects that the latest climate change scenarios may have on this transfer, one of the most-important ones in southern Europe. In the first place, the possible alterations of the water cycle of the donor basin were estimated. To do this, a hydrological model was calibrated. Then, with this model, three climatic scenarios were simulated, one without climate change and two projections under climate change (Representative Concentration Pathways 4.5 (RCP 4.5) and RCP 8.5). The results of these three hydrological modelling scenarios were used to determine the possible flows that could be transferred from the Tagus River Basin to the Segura River Basin, by simulating the water resources exploitation system of the Tagus headwaters. The calibrated hydrological model predicts, for the simulated climate change scenarios, important reductions in the snowfalls and snow covers, the recharge of aquifers and the available water resources. So, the headwaters of the Tagus River Basin would lose part of its natural capacity for regulation. These changes in the water cycle for the climate change scenarios used would imply a reduction of around 70-79% in the possible flows that could be transferred to the Segura Basin, with respect to a scenario without climate change. The loss of water resources for the Segura River Basin would mean, if no alternative measures were taken, an economic loss of 380-425 million € per year, due principally to decreased agricultural production.

## 1 Introduction

Currently, there are practically no doubts in the scientific community that the Earth is suffering climate change (CC), and that this is due to the anthropic action of greenhouse gas emissions (IPCC, 2014). At the global level, general circulation models

predict a warming of the planet of about 2 °C for the year 2050, which will cause a reduction in accumulated ice masses and a rise in the sea level (IPCC, 2014). These changes in the natural environment, which are already causing alterations in the available resources, have a clear socio-economic repercussion: decreases in fish stocks, increases in energy consumption, changes in the availability of water resources and land degradation due to erosion. In areas with greater risk and with low capacity for adaptation, the consequences of CC could become critical, the emigration of their population being the only viable solution (Jha et al., 2018).

In the context of CC, water as a resource plays a fundamental role, since - as well as being a basic environmental asset - it is key to human survival and well-being. In general terms, an increase in rainfall in humid areas and a decrease in arid and semi-arid ones are predicted. This situation would be accompanied by an increase in the frequency and intensity of extreme events (droughts and floods), so in areas where already there are shortages of water resources the current situation would be exacerbated (IPCC, 2014). For areas close to polar regions or mountainous areas, the increase in temperature would reduce the precipitation that falls as snow (Szczypta et al., 2015), as well as the volume of ice in the glaciers and the snow cover on the summits (Bajracharya et al., 2018). As a consequence, the fluvial regime for this type of river basin would be modified (Morán-Tejeda et al., 2014), with an increase in the risk of floods and the loss of the part of their natural regulatory capacity provided by the ice and snow covers (Shevnina et al., 2017). Thus, in areas where the water reserves derived from snow covers are used during the summer, new reservoirs would have to be built in order to replace the loss of the natural regulation capacity (Özdoğan, 2011). Similarly, in areas with important aquifers, part of the natural regulatory capacity could also be lost, since changes in precipitation patterns would affect recharge rates (Smerdon, 2017). Indeed, a higher intensity of rainfall would favour surface runoff to the detriment of infiltration (Pulido-Velazquez et al., 2014). In short, the CC predicted for many areas of the planet will suppose an increase in temperature together with a greater availability of surface water - which would increase the water evapotranspiration, accelerating the water cycle and reducing the available water resources that could be used (Wang et al., 2013).

The environmental and social repercussions of these physical effects are being studied from multiple perspectives (Olmstead, 2014; World Bank Group, 2016). For example, there is work related to water quality (Molina-Navarro et al., 2014), the effects on ecosystems and the services they provide (Warziniack et al., 2018) and the impact on the food security (Tumushabe, 2018) and water security (Flörke et al., 2018) of the population. Although, the majority of the studies deal with the impacts on the economic activities which are more sensitive to the availability of water resources, such as agriculture (Meza et al., 2012), urban supply (Díaz et al., 2017) or the hydroelectric sector (Solaun and Cerdá, 2017).

Water resources transfers between basins (inter-basin water transfer: IBWT) are instruments of water resources allocation that, despite the controversy they sometimes provoke, can play an important role in mitigating the effects of CC in many areas of the world (Shrestha et al., 2017). The IBWTs can be an alternative source of supply to basins affected by a decrease in their available resources and/or an increase in their water use demands (Zhang et al., 2015). In turn, the quantity of available water in the donor basins can change significantly, which makes CC a determining factor that must be analysed when assessing the potential or vulnerability of the IBWT (Zhang et al., 2018). Although there are some works that specifically studied the effects of CC on IBWTs, basically the focus has been on changes in the fluvial regimes in donor basins (Zhang et al., 2012; Li et al.,

2015). Moreover, there are scarce examples in the specialised literature of the analysis of the effects of CC within a framework of integrated water resources management (Onagi, 2016). In addition, for an adequate comprehensive study of the effects of CC in an IBWT, and to produce operational indicators for water management plans (Giupponi and Gain, 2017), including the management of water trade (Kahil et al., 2015), it is also necessary to consider the effects in the receiving basin.

The Tagus-Segura Aqueduct (TSA) is one of the most-important IBWT projects in southern Europe. This hydraulic infrastructure, in operation since 1979, transfers flows from the Tagus Headwaters River Basin (THRB) to the Segura River Basin (SRB). The destination of the volumes transferred, which are variable depending on the available water resources in the donor basin, is basically irrigation, but also urban and tourism uses (Grindlay et al., 2011). The regional models of CC forecast, for both basins, an increase in temperature together with a significant diminution in rainfall, so that a decrease in the available

water resources in both is foreseen (CEDEX, 2011a). In addition, the THRB is located in a high, mountainous area where snowfalls are frequent and which extends over important karst aquifers. Then, it is expected that both the precipitation that falls as snow and the aquifer recharge would be reduced. Although there is work in which this problem is explicitly described, with proposals to mitigate the decrease in TSA flows due to CC (Morote et al., 2017), no specific modelling of climate scenarios has been made, neither from a hydrological perspective, to determine the water balance, nor by simulation of the water resources

exploitation system.

The overall objective of this work was to determine the hypothetical effects that CC may have on the operation of the TSA. So, the transferable flows were estimated considering explicitly the operating rule of this IBWT - which, basically, is based on the available water storage in the main two reservoirs of the donor basin concerned. For this, first of all, the effects on the water cycle of the donor basin were evaluated by means of hydrological modelling in which the precipitation that falls

as snow was included. Then, the historical climate data together with two CC scenarios of the fifth assessment report (AR5) of the Intergovernmental Panel on Climate Change (IPCC, 2013) were recreated in order to analyse the possible alterations in the fluvial regime of the THRB. The results of these three hydrological models were the inputs of the subsequent three simulations of the Tagus Headwaters Water Resources Exploitation System (THWRES), which provided a prediction of the flows that could be transferred to the SRB within a framework of integrated water resources management. Additionally, as

another novel contribution of this work, the socioeconomic impacts produced by the climate change on these transferred flows were assessed. Finally, note that the complete methodology was developed by Open Source tools and by free software for scientific community, which facilitates the reproducibility of the work.

## 2    The water resources exploitation system of the Tagus Headwaters River Basin

The THRB covers an area of 7000 $\text{km}^2$ and is located in the middle of the Iberian Peninsula (1). It extends over a high,

mountainous area having a Continental-Mediterranean (CHJ, 2016) climate with a marked seasonality between the summer (June-September) and winter months (December-March) (Lorenzo-Lacruz et al., 2010; Molina-Navarro et al., 2014). The average annual precipitation is 620 $\text{mm}$ and the minimum values occur in summer (June-August). While the average annual temperature is 11 $^\circ\text{C}$, in the coldest months there are values less than zero (November-April), so snowfalls are frequent at the

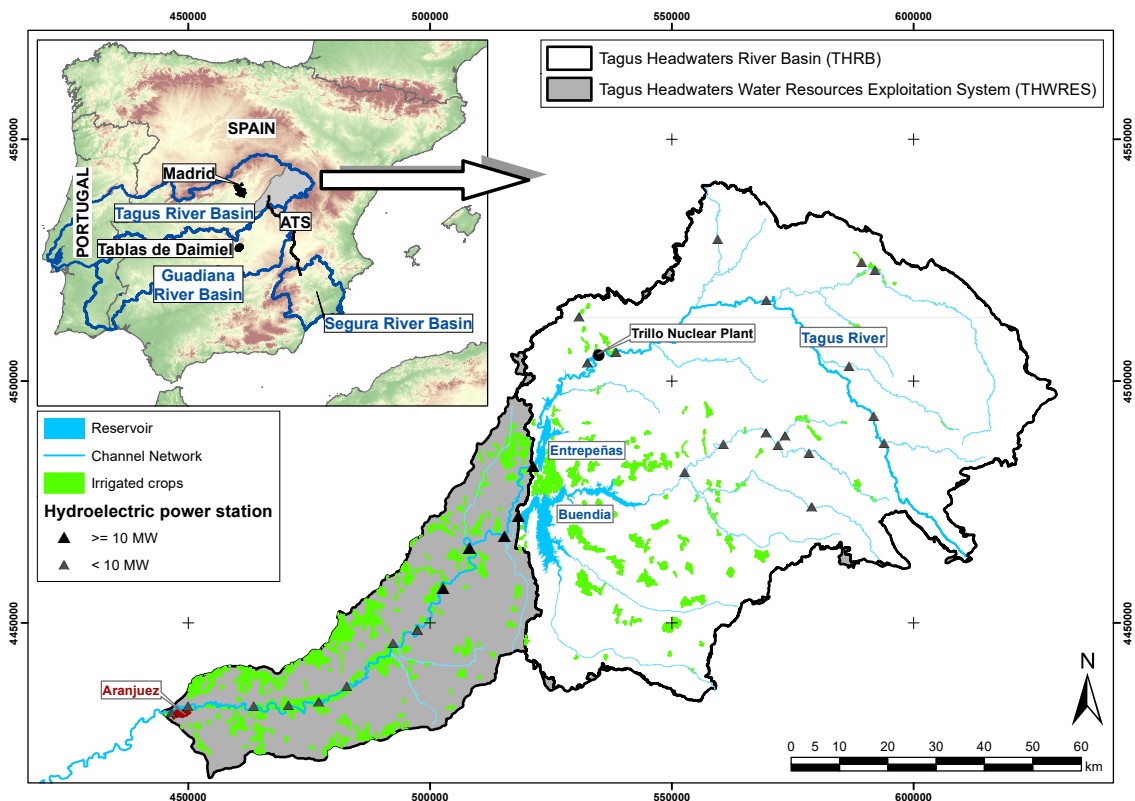

**Figure 1.** The Tagus Headwaters River Basin (THRB) and Tagus Headwaters Water Resources Exploitation System (THWRES): the main physical characteristics and water uses (including the TSA). Source data: CHT (2018).

higher altitudes (Lobanova et al., 2016). Moreover, much of the THRB extends over karst aquifers, meaning that groundwater exerts an important influence on the surface flows that circulate along the main river streams (Pellicer-Martínez et al., 2015).

Regarding the water uses within the THRB (urban, industrial and irrigation), their water necessities represent a very-low percentage of the available water resources (around $1000 \cdot 10^6 \, \text{m}^3 \, \text{yr}^{-1}$, on average, in the last 70 years): this area has a low population density, is not conducive to agriculture and its industrial facilities (hydroelectric power stations and an important thermonuclear power station) do not consume much water.

The water resources generated within the THRB are fundamental for the water uses located downstream of the Entrepeñas and Buendía reservoirs (EBR): irrigated agriculture, urban supply (including the city of Madrid), generation of electric power and maintenance of environmental flows until the city of Aranjuez. Moreover, a large part of these water resources (up to $650 \cdot 10^6 \, \text{m}^3 \, \text{yr}^{-1}$) is susceptible to being transferred to the neighbouring Guadiana River Basin and, further away, to the SRB. The former can receive up to $50 \cdot 10^6 \, \text{m}^3 \, \text{yr}^{-1}$ (BOE, 2014, 2015), of which $20 \cdot 10^6 \, \text{m}^3$ are for the maintenance of the wetland of Tablas de Daimiel, and $30 \cdot 10^6 \, \text{m}^3$ are for urban supply to the populations located at the upper of the Guadiana River Basin (CHG, 2016). The SRB can receive up to $600 \cdot 10^6 \, \text{m}^3 \, \text{yr}^{-1}$ (gross volume), the maximum monthly flow being $68 \cdot 10^6 \, \text{m}^3$. This

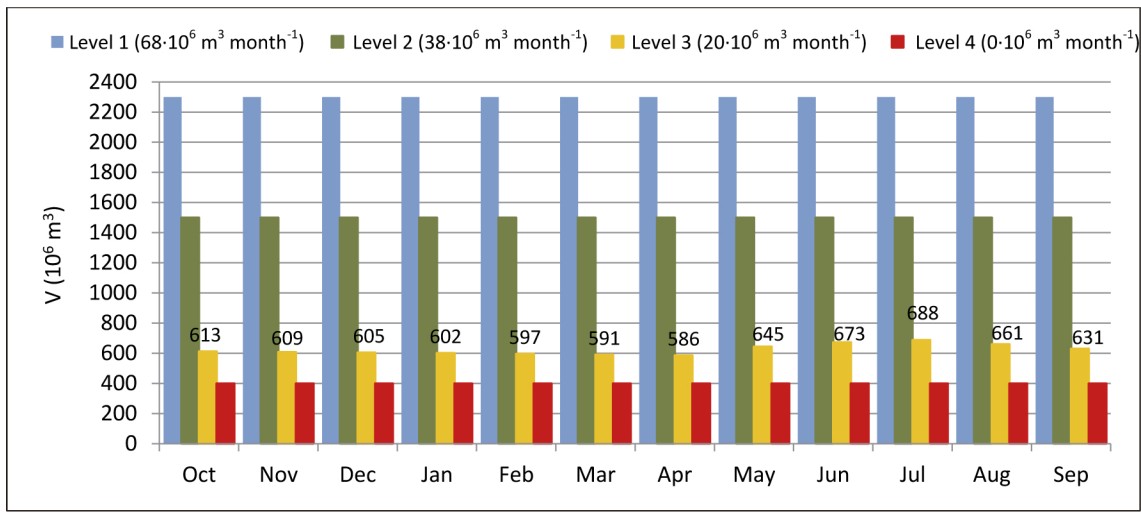

**Figure 2.** Operating rule of the TSA. Based on BOE (2014, 2015).

IBWT is managed by a complex operating rule that gives priority to the water uses in the Tagus river basin and basically depends on the volume stored in the EBR, which have a total storage capacity of $2494 \cdot 10^6 \, \text{m}^3$. The operating rule (Fig. 2) consists of two conditioning factors (BOE, 2014, 2015). The first restricts the maximum volume that can be transferred in each hydrological year (October-September) to $650 \cdot 10^6 \, \text{m}^3$. The second establishes the transferable flows for each month according to four levels created from two variables: $V_{acu}$, the accumulated volume stored in the EBR at the beginning of the month, and $A_{acu}$, the accumulated volume of the flows that entered into the EBR in the previous 12 months. The four levels are:

– Level 4. When $V_{acu}$ is lower than $400 \cdot 10^6 \, \text{m}^3$. Transfers are not allowed ($Q_{trans} = 0 \, \text{m}^3 \, \text{month}^{-1}$).

– Level 3. When $V_{acu}$ is between $400 \cdot 10^6 \, \text{m}^3$ and the values indicated in Fig 2, which vary between $586 \cdot 10^6 \, \text{m}^3$ and $688 \cdot 10^6 \, \text{m}^3$ depending on the month. Transfers ($Q_{trans}$) of $20 \cdot 10^6 \, \text{m}^3 \, \text{month}^{-1}$ are allowed.

– Level 2. When $V_{acu}$ is between the volumes established in Level 3 and $1500 \cdot 10^6 \, \text{m}^3$, and in addition $A_{acu}$ is lower than $1000 \cdot 10^6 \, \text{m}^3$. Transfers ($Q_{trans}$) of $38 \cdot 10^6 \, \text{m}^3 \, \text{month}^{-1}$ are allowed.

– Level 1. When $V_{acu}$ is equal to or greater than $1500 \cdot 10^6 \, \text{m}^3$, or $A_{acu}$ is equal to or greater than $1000 \cdot 10^6 \, \text{m}^3$. Transfers ($Q_{trans}$) of $68 \cdot 10^6 \, \text{m}^3 \, \text{month}^{-1}$ are allowed.

## 3    Methodology

The methodology applied to determine the maximum monthly volumes that can be transferred from the THRB to the SRB was structured in two stages (Fig. 3).

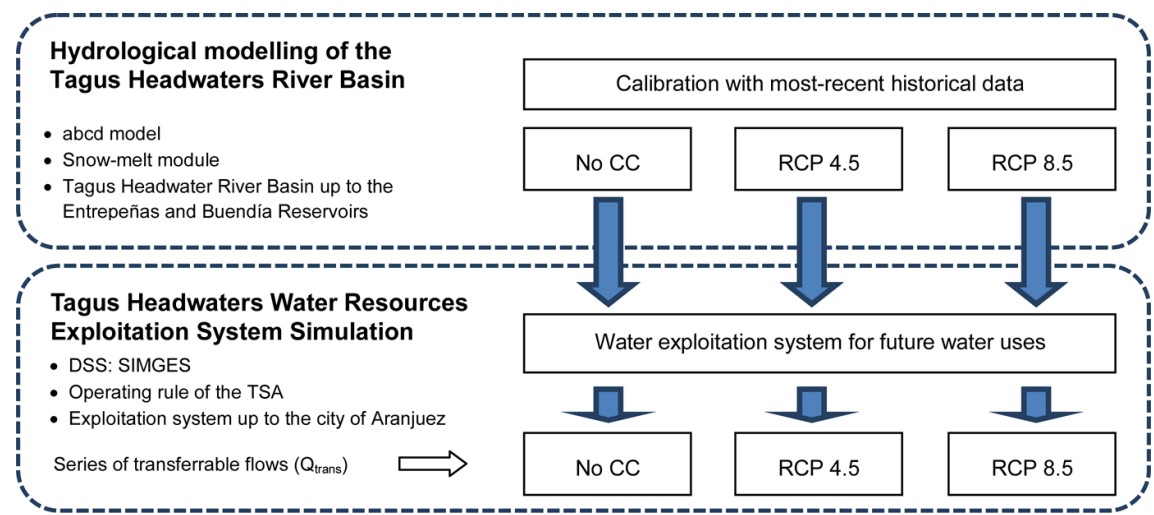

**Figure 3.** Methodological framework.

The first stage consisted of modelling the hydrology of the THRB until the EBR. For that, a hydrological model was calibrated for the most-recent observed flows. Then, with the calibrated model, three scenarios were recreated. In the first one the historical climate series were used (No CC) without climatic correction coefficients. In the others, the data from two characteristic CC scenarios of AR5 (Representative Concentration Pathways 4.5 (RCP 4.5) and RCP 8.5) were used (IPCC, 2013). In the first (RCP 4.5), $CO_2$ emissions increase in the future, until in 2050 they stabilise (stabilisation scenario). While in the second (RCP 8.5), a continuous and greater increase in emissions of $CO_2$ is assumed (scenario of very-high emissions).

The second stage consisted of simulating the THWRES by means of a decision support system (DSS). This simulation incorporated the future water uses contemplated by the water management board. As water uses downstream of the EBR have priority over possible transferable flows (CHT, 2015), they were included in this simulation.

The methodological framework stages were developed with Open Source tools. QGIS (QGIS Development Team, 2016) was used in the data processing of spatial information, R was employed for data analysis and hydrological modeling (R Core Team, 2016), whereas the DSS SIMGES was used for the simulation of the water resources exploitation system (Pedro-Monzonís et al., 2016a).

Finally, once the series of transferrable flows were calculated, as a complementary goal of this work, an assessment of the socioeconomic consequences that climate change effects have on the main destiny of these flows, the SRB, was made.

## 3.1 Hydrological modelling

### 3.1.1 abcd water balance model with snow-melt module

The hydrological modelling was carried out using the abcd water balance model (Thomas, 1981). It was applied in a semi-distributed manner (Pellicer-Martínez and Martínez-Paz, 2014), allowing the use of all the gauging stations in the calibration

(and validation) in order to maintain the spatial heterogeneity that defines the parameters and variables. This conceptual model was improved by taking into account the hydrological processes of snow and melting. This water balance model and this structure were selected in order to facilitate the understanding of the developed process, allowing the potential reproducibility of the work.

The hydrological modelling, whose scheme is shown in Figure 4, began with the snow-melt module proposed by Xu and Singh (1998). This module recreates the hydrological processes of precipitation as snow ($S_n$), snow accumulation on the summits ($S_{np}$) and snow melt ($S_m$). One equation, with a parameter that depends on the temperature, establishes which part of the precipitation (P) occurs as rain ($R_f$) and which part occurs as snow ($S_n$). Snow is accumulated in a storage called snowpack ($S_{np}$). Then, another equation controlled by one parameter, which also depends on the temperature, establishes the snow melt

($S_m$) of the snowpack when the temperature increases. In the modelling, the melting snow ends up forming part of the storage that represents the soil moisture (S). This snow-melt module has been used in previous work; for example, by Li et al. (2011, 2013) in basins of China, and by Pellicer-Martínez and Martínez-Paz (2015) for the Segura headwaters river basin.

    The modelling continued with the incorporation of rainfall ($R_f$) and melting snow ($S_m$) into the abcd model. This water balance model simplifies the hydrological cycle in two storages, one that simulates the soil moisture balance (S) and another

that represents the groundwater storage (G). The model has four parameters - a, b, c and d (Thomas, 1981) - that give the model its name. The parameters "a" and "b" manage the soil moisture balance (S), establishing evapotranspiration ($E_T$) and the water susceptible to being lost by surface run-off and/or percolation ($Q_s + \Delta G$). The third parameter "c" identifies percolation ($\Delta G$) towards aquifers (G) and surface run-off ($Q_s$). The fourth parameter "d" determines the discharge from the aquifer ($Q_g$). The output variables of the model are evapotranspiration ($E_T$) and surface run-off (Q).

**3.1.2   Calibration-validation process**

The parameters were calculated by a cascading calibration process (Xue et al., 2016), which consists of determining the parameter values from upstream to downstream. In other words, once the model's parameters are established for upstream catchments, their values become input data in the calibration process of the downstream catchments. The split-sample test, proposed by Klemeš (1986), is carried out by splitting the data series of observed flows into two periods: calibration and

validation.

    The objective function used in the calibration is the Nash-Sutcliffe efficiency criterion ($E_{NS}$). This function quantifies the goodness of fit of the model in order to evaluate the performance of the variables selected for study (Nash and Sutcliffe, 1970), which are the flows that go out from each catchment (Q). The $E_{NS}$ varies between $]-\infty, 1]$ and the closer its value is to 1, the better is the performance of the model. The calibration is developed automatically with the algorithm Shuffled Complex

Evolution Method (SCE-UA) (Duan et al., 1994; Skøien et al., 2014). Once the semi-distributed model has been calibrated, another three metrics that evaluate the performance of the model, comparing observed with simulated flows, are calculated: the determination coefficient ($R^2$), the percentage of the bias ($P_{BIAS}$) and the root mean square error ($E_{RMS}$) (Gupta and Kling, 2011).

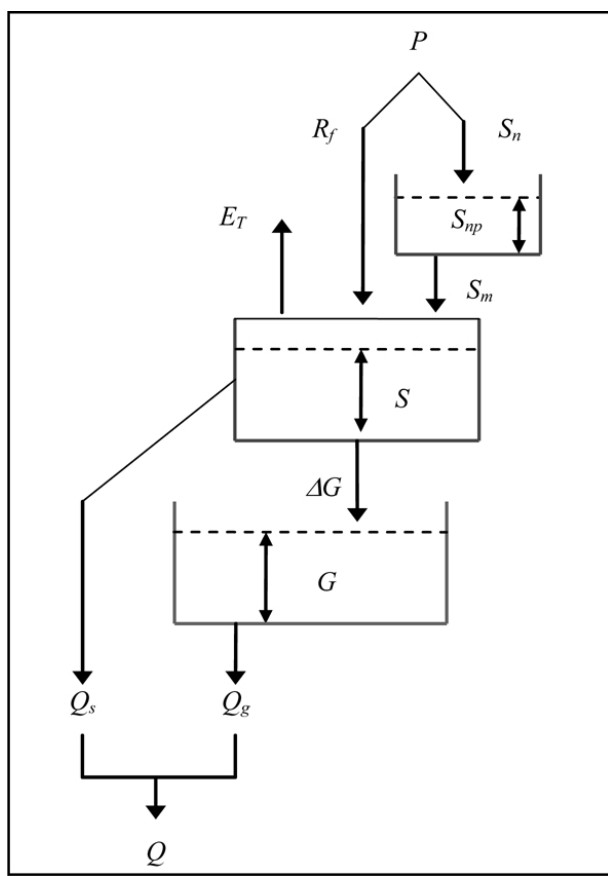

**Figure 4.** Conceptual scheme of the abcd model.

## 3.2 Simulation of the water resources exploitation system

The operation of the water exploitation systems for a specific scenario is usually evaluated by a DSS which simulates the part of the water cycle that is anthropically modified. These systems represent the main hydraulic network of an area, generally a river basin, together with its natural and artificial storages (rivers, reservoirs, aquifers, canals, among others), establishing the water
5  uses in them (Pulido-Velazquez et al., 2013). In this hydraulic network, series of flows are introduced, generally in their natural regime. Then, the DSS simulates this kind of system in an integrated manner, fulfilling the priority criteria among the different water uses and the pre-established operating rules (Pedro-Monzonís et al., 2016a). The DSS provides, as results, flow and volume series related to the main fluxes of the hydraulic network (supplies to uses, water consumptions, returns, evaporation in the reservoirs, etc.). Therefore, it is able to estimate the water uses that are not completely met, which are generally those
10  with lower priority and/or those whose spatial location in the hydraulic network does not make it possible to guarantee always their supply (Chavez-Jimenez et al., 2015). For example, water uses located upstream of a reservoir have less guarantee than those located downstream.

There are different DSSs for water exploitation systems (Zare et al., 2017). But, as it was advanced in the methodology section, in this work the model applied is the simulation module of AQUATOOL called SIMGES ((Pedro-Monzonís et al., 2016b), which is one of the models applied most in Spanish river basins, as well as in other countries (Chile, Italy, Morocco, etc.). SIMGES simulates the water exploitation system on a monthly basis in a conservative flow network, seeking a compatible solution that accomplishes the defined constraints (Pedro-Monzonís et al., 2016b). This DSS was selected since it is able to reproduce complex operating rules such as the TSA operating rule.

### 3.3 Socioeconomic impacts at Segura River Basin (SRB)

The precise quantification of the socioeconomic impact of reductions in the volume of water transferred via the TSA would require new integral simulations of the exploitation system of the receiving basin (SRB), which exceed the scope of this work. However, an initial quantification of this impact has been made, based on the work of Martínez-Paz et al. (2016), where supply failures were assigned an economic value in irrigation in the SRB, and of Martínez-Paz and Pellicer-Martínez (2018), who estimated the economic value of the risk associated with droughts in the Region of Murcia. But, given the order of priority of allocation among the water uses, irrigation would suffer the full brunt of any supply deficit. The almost 270000 ha of irrigated land in the SRB has a net demand of $1363 \cdot 10^6 \, \mathrm{m}^3 \, \mathrm{yr}^{-1}$ (CHS, 2015), a large part of which is supplied by the TSA. The whole irrigated area in the SRB is divided into seven irrigation zones (IZs); for each of them the water demand curve is estimated from a linear programming model that optimises the Gross Value Added (GVA) by the optimal cultivation plans according to the water supply. This crop programming includes the irrigation situations of woody crop maintenance, the change from irrigated to rainfed crops and the abandonment of irrigation plots, as well as the impact on employment.

The modelling of the optimal crop plan for each IZ is determined by the following objective function (1) that maximizes the GVA:

$$\max \sum_i \left( Y_i \cdot P_i - C_i \right) \cdot L_i \tag{1}$$

Where $i$ denotes crop activities under different management options, $Y_i$ is the yield of each crop $i$, $P_i$ the price received by the farmer, $C_i$ the direct costs of production per unit area, and $L_i$ is the area dedicated to each activity.

The objective function is subject to the following constraints (2-7):

$$\sum_i L_i \leq L_T \tag{2}$$

$$\sum_i q_i \cdot L_i \leq Q_T \tag{3}$$

$$L_i^R + L_i^S = L_i^E \tag{4}$$

$$L_i^R + L_i^M + L_i^P = L_i^E \tag{5}$$

$$L_i^G \leq L_i^E \tag{6}$$

$$\sum_i lf_i \cdot L_i \leq LF_T \tag{7}$$

$L_T$ is the total available surface irrigable in the IZ; $q_i$ is the water requirement of each crop per unit area and $Q_T$ is the availability of water for the entire campaign in the IZ; $L_i^R$ is the surface of irrigated woody crops; $L_i^S$ is the irrigable surface that goes to rainfed; $L_i^M$ is the irrigable surface of woody crops under maintenance irrigation; $L_i^P$ is the surface of the irrigation plots abandoned; $L_i^G$ is the surface of existing irrigated greenhouses; $L_i^E$ is the existing area of each activity in the basin in the reference year; $lf_i$ are the labor requirements for each crop, and $LF_T$ is the availability of agricultural labor in the IZ.

The first constraint (2) prevents that each unit of demand (IZ) cultivates more area than the available net irrigable area. The following constraint (3) represents the limitation of water availability for each IZ. The set of constraints (4), (5) and (6) allow simulating specific management options to certain crop groups. The constraint (4) fixes the total area of woody crops such as almond, olive and wine, distributed between irrigated and rainfed depending on the availability of water. The constraint (5) represents citrus and fruit trees, whose total area is equal to the area actually irrigated plus, in situations of scarcity of resources, the surface under maintenance irrigation and/or loss of trees because of not being able to perform the minimum maintenance irrigation. The constraint (6) sets the maximum available area for greenhouse crops in the reference year. Finally, the constraint (7) represents the limitation of the available labor for each IZ. The program used allows estimating the gross margin generated under different water availability assumptions, as well as deriving water demand curves and the marginal value of this resource (Griffin, 2006).

The necessary data to characterize the technical coefficients of each IZ have been obtained from the sources indicated in Martínez-Paz et al. (2016) and of Martínez-Paz and Pellicer-Martínez (2018), updating all the economic figures to € of 2017. The program was solved for each IZ and for the three climatic scenarios (No CC, RCP 4.5 and RCP 8.5), so the availability of water in each of them changes ($Q_T$). The differences in the average volume transferred to the SRB in each CC scenario (RCP 4.5 and RCP 8.5) calculated with SIMGES were distributed proportionally for each IZ, taking as reference the volume transferred in the scenario without CC. Thereby, the socioeconomic impact due to changes in the availability of water was obtained, *ceteris paribus* the rest of the parameters of the model. Finally, the comparison among the results obtained from GVA and employment for each scenario are presented in the Discussion section.

## 4 Data source

### 4.1 Hydrological modelling of the Tagus Headwaters River Basin

The Digital Elevation Model (DEM) employed has a 25-m resolution and is available on the website of the National Geographic Institute of Spain (www.cnig.es). This DEM was used as an auxiliary variable in the interpolation models of the climatic variables, and to delimit the main streams and catchments using D8 algorithm (O'Callaghan and Mark, 1984). The locations of the 12 gauging stations, which have observed flows in the same period, were used to establish the outlet points of the 12 catchments in which the THRB was divided (Fig. 5). The data series of the observed flows are available in the gauging yearbook of the Official Gauging Station Network of Spain (MITECO, 2018) and cover the period from September 1985 to December 2009. These observed flows have been previously naturalized to be used in the calibration-validation process (Wurbs, 2006). For that, the main human alterations located upstream of each gauging station were undone: regulation and evaporation in the reservoirs, as well as the derivations for urban, agricultural and industrial uses (also the returns of these uses were considered). As available data for the 12 gauging stations exist for the same period, it is possible to calibrate the model jointly for all the catchments. Since the objective of this work is to obtain a statistically significant calibration and data for its testing, each series of observed flows was divided into two periods (Klemeš, 1986), the first (from September 1985 to July 1995) being used in the validation and the second (from August 1995 to December 2009) being used in the calibration. Thus, the parameters to be used for the CC projections were determined with the most-recent data.

The historical climate series used in the calibration-validation comprise the period from September 1980 to December 2009. Thus, the five years before the observed flows series were used to warming up the hydrological model. Two information sources were used to obtain the monthly climatic series. The first was the dataset Spain02v5 (Herrera et al., 2016), the historical series used for the calibration-validation of the hydrological model and for its simulation in the scenario without climate change (No CC). For the No CC scenario, these series of data were extended from October 1940 to September 2010 (seventy consecutive years), and they were used in the simulation without climatic correction coefficients. The advantages of using Spain02v5 are that the daily series of precipitation and temperature are refined (without outliers and/or in homogeneities), and that they include the spatial variability of the climatic variables in a grid with a 12.5-km resolution (Fig. 5). Since these data are daily, they were aggregated on a monthly basis to apply them in the model. The second source data was the State Meteorological Agency of Spain (AEMET) and was used for the two CC scenarios of AR5 (IPCC, 2013). This source (AEMET) provides the regionalised projections of 27 models (13 for RCP 4.5 and 14 for RCP 8.5), using the statistical method of analogues (Amblar-Francés et al., 2017). The daily temperature and precipitation series of the six thermometric and 48 precipitation stations closest to the THRB were used (Fig. 5). These daily data were also aggregated to monthly series. Next, the reference historical data series of each model were compared with Spain02v5 data, using as a control period the 1971-2005 interval. The 10 models that best fitted for both temperature and precipitation were assembled using the Simple Average Forecast Combination (SA), and Bias-Corrected Eigenvector Forecast Combination (EIG2) (Hsiao and Wan, 2014), available in the R-CRAN package GeomComb (Weiss and Roetzer, 2016). Then, both ensembles were also compared with Spain02v5 data using the same control period (1971-2005). Finally, the series obtained by EIG2 were used, with a lower prediction error compared to the rest of series. For example, the

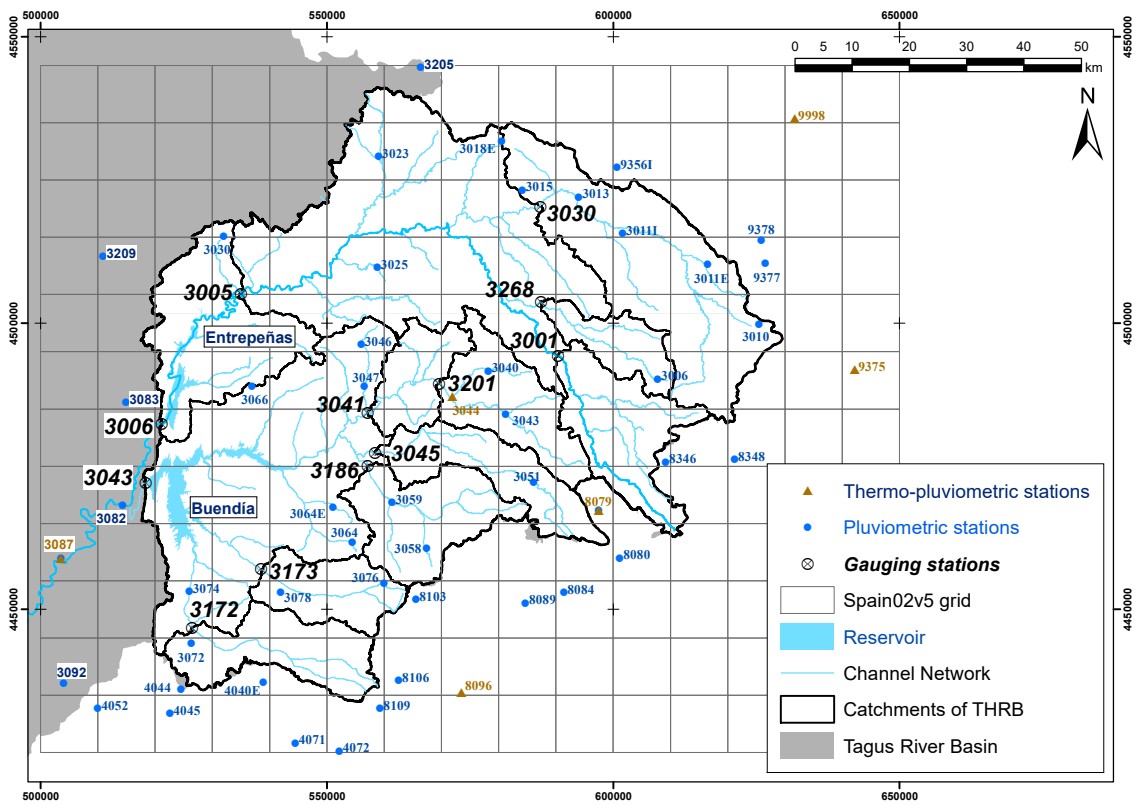

**Figure 5.** Spatial distribution of climate stations with AR5 information and the grid obtained from Spain02v5.

ENS values obtained with EIG2 for the temperatures were 0.87, while in the separate models it never exceeded 0.77. For the precipitations, the ENS value was 0.30, while in the models it was always lower than 0. In addition, another advantage of EIG2 method is that it allows to correct the bias produced by predictive models. The period used in the simulation of CC scenarios (RCP 4.5 and 8.5) is from October 2020 to September 2090 (also seventy consecutive years).

5    Based on the average monthly temperature data of the three climatic scenarios, the potential evapotranspiration series were estimated using the Thornthwaite method (Thornthwaite, 1948; Gomariz-Castillo et al., 2018). As this method tends to under-estimate the potential evapotranspiration, the series generated were corrected from a linear regression between the estimated series and those used by the SIMPA hydrological model (BOE, 2007). These series used by SIMPA have already regionalized for the Iberian Peninsula based on the Penman-Monteith method (Allen et al., 1998), correcting the underestimation of the 10    Thornthwaite method.

Once the monthly series of precipitation, temperature and potential evapotranspiration had been calculated, they were spatially interpolated on the cells using the Thin-Plate Splines method (Wahba, 1990), which is based on local interpolation from polynomials. This method turns out to be relatively robust against non-compliance with the statistical assumptions necessary in methods such as Kriging, being used with good results for the interpolation of climatic variables such as rainfall (Hutchinson,

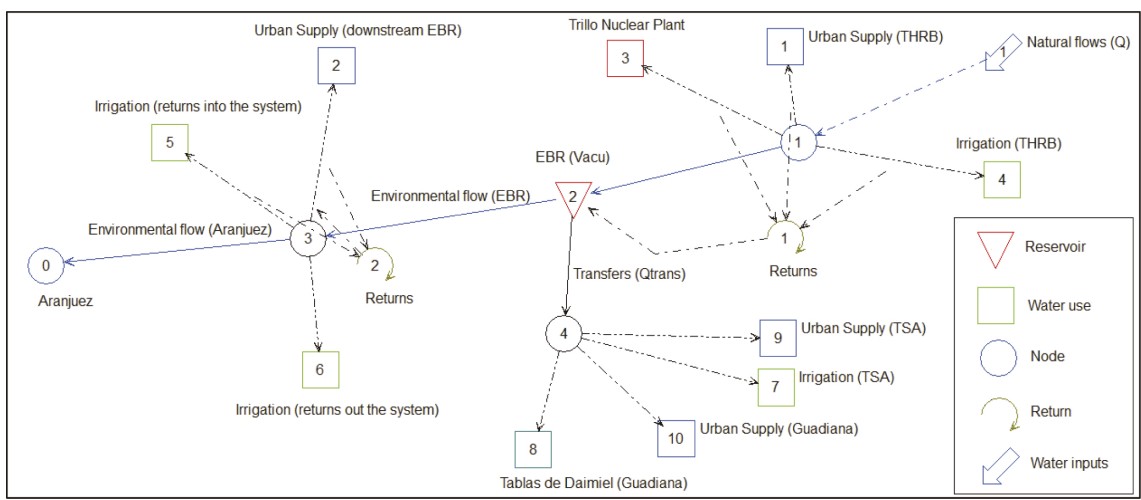

**Figure 6.** Scheme of the water resources exploitation system of the Tagus Headwaters, as far as Aranjuez.

1995) and temperatures (McKenney et al., 2006). Finally, each catchment was assigned the average value of the cells over which it extend.

## 4.2 Water resources exploitation system of the Tagus Headwaters Basin (THWRES)

The THWRES covers the basin upstream of the EBR and the water uses located in the basin downstream of these reservoirs,
up to the city of Aranjuez (Fig. 1 and Fig. 6). In its design, the possible future water uses for the years 2016, 2021 and 2033 contemplated by the water management board (CHT, 2015) were analysed. And, as there are no relevant differences between them, the uses predicted for the year 2033 were taken as the reference for the three climatic scenarios (No CC, RCP 4.5 and RCP 8.5) (Table 1). In addition, as the available water resources in the catchments downstream of the EBR are so low with respect to those generated in the EBR drainage basin, they were neglected in the water resources exploitation system
simulation.

   Regarding water uses, the majority of the urban and irrigation uses are concentrated downstream of the EBR, requiring more than 85% of the total volume demanded. For urban use, a water return of 80% was considered with respect to the volume supplied. In 2033 irrigation will have a low water return into the system ($< 1\%$) due to improvements in irrigation facilities and in the application systems which will be used, so this low return was implemented in the DSS. The main industrial use of water
occurs at the Trillo Nuclear Plant. This industrial use requires a constant flow over time that returns 46% of the water supplied. Finally, an environmental flow of around 11 m$^3$ s$^{-1}$ must circulate in the Tagus River from the EBR to the city of Aranjuez. The order of priority in the water resources allocation among the different water uses is: urban supplies, Trillo Nuclear Plant and irrigation. Once the water uses in the THWRES have been supplied, the possibility of authorising transfers is evaluated (BOE, 2014, 2015). Among the four uses of the transfer there is no stipulated clear criterion in the operating rule that governs
it, since sometimes it is done discretionally depending on the needs or level of urgency of the uses. In fact, the transfer to

**Table 1.** Main water uses in the Tagus Headwaters Water Resources Exploitation System (THWRES) (unit: $10^6 \, \mathrm{m}^3 \, \mathrm{yr}^{-1}$).

| Water uses in THWRES | Annual water demanded |
|---|---|
| Environmental flow in the Tagus River downstream of the EBR | 327.0 |
| Environmental flow in the Tagus River in the city of Aranjuez | 258.0 |
| Urban water use | 35.5 |
| Irrigation water use | 159.1 |
| Industrial water use (Trillo Nuclear Plant) | 37.8 |
| Transfers Segura (urban and irrigation supplies) | 600.0 |
| Transfers Guadiana (Tablas de Daimiel) | 20.0 |
| Transfers Guadiana (urban supply) | 30.0 |

Tablas de Daimiel has occurred only once, to avoid serious damage to this wetland. Therefore, in order to accomplish the general criteria of the Spanish legislation, the order of priority followed is: urban supply to Segura (TSA), urban supply to the populations located at the upper of the Guadiana River Basin, irrigation supply to Segura (TSA) and Tablas de Daimiel.

## 5 Results

### 5.1 Climate change effects on the hydrology of the Tagus Headwaters River Basin

The values of the criterion coefficients calculated in the hydrological modelling show that the model employed reproduced properly the surface flows in the THRB in the calibration period: high values of $E_{NS}$ and $R^2$, together with low relative errors ($E_{RMS}$) and volume errors ($P_{BIAS}$). However, in the validation period, there are some low values for the goodness of fit coefficients calculated, indicating that the results of these catchments have greater uncertainty. These results can be explained by the fact that the validation period was used just after the warming up process. Thereby, it could cause that, in some catchments, the warming up can extend to a part of the validation period. This would entail that the calibration process used a part of the validation to adjust the initial parameters, obtaining worse adjustments in the validation of the model. However, it is important to highlight that the parameters used in the simulations are adjusted with the more recent data, providing a good performance of the surface flows in the THRB. In addition, the best performance in calibration corresponded to the outlets of the catchments of Entrepeñas and Buendía, which were the flows used as the input in the subsequent simulation of the water resources exploitation system. In fact, both catchments had NSE values around 0.80 and low $P_{BIAS}$ (Table 2).

The simulation of the historical climate series of 1940-2010 with the calibrated model provided an average annual resource of $954.6 \cdot 10^6 \, \mathrm{m}^3 \, \mathrm{yr}^{-1}$ (Q). This series was temporally moved to the 2020-2090 time period in order to reproduce a future climate scenario without climate change (No CC). The simulations for climate change scenarios RCP 4.5 and RCP 8.5 with the same calibrated model indicated that the THRB could suffer a considerable loss of its natural water resources. The RCP 4.5 scenario forecasted a value of $575.6 \cdot 10^6 \, \mathrm{m}^3 \, \mathrm{yr}^{-1}$, representing a decrease of 39.7%, while the RCP 8.5 scenario predicted a 46.6%

**Table 2.** Results of the calibration-validation process for the 12 gauging stations in the hydrological modelling.

| Gauging stations | | Calibration | | | | Validation | | | |
|---|---|---|---|---|---|---|---|---|---|
| Code | Name | $R^2$ | $E_{NS}$ | $P_{BIAS}$ | $E_{RMS}$ | $R^2$ | $E_{NS}$ | $P_{BIAS}$ | $E_{RMS}$ |
| 3001 | Peralejos de las Truchas | 0.76 | 0.74 | 16.60 | 6.11 | 0.51 | 0.43 | -2.90 | 6.16 |
| 3268 | Taravillas | 0.57 | 0.57 | 0.70 | 1.34 | 0.43 | 0.19 | -32.20 | 2.51 |
| 3030 | Ventosa | 0.86 | 0.86 | -2.70 | 2.90 | 0.61 | 0.34 | -5.20 | 4.07 |
| 3005 | Trillo | 0.51 | 0.50 | 1.10 | 0.79 | 0.47 | 0.45 | -8.10 | 0.40 |
| 3045 | Priego Escabas | 0.67 | 0.67 | 3.50 | 0.60 | 0.10 | -0.36 | 19.30 | 0.49 |
| 3172 | Huete | 0.64 | 0.64 | -1.90 | 0.90 | 0.38 | 0.37 | -10.20 | 1.33 |
| 3173 | La Peraleja | 0.82 | 0.82 | -1.30 | 2.72 | 0.31 | 0.15 | 34.80 | 4.95 |
| 3186 | Priego Trabaque | 0.70 | 0.70 | -2.10 | 1.30 | 0.69 | 0.57 | -16.20 | 1.14 |
| 3201 | Molino de Chincha | 0.79 | 0.78 | 11.00 | 15.14 | 0.61 | 0.46 | -9.50 | 16.11 |
| 3041 | Alcantud | 0.72 | 0.71 | -8.60 | 7.69 | 0.53 | 0.35 | -15.70 | 6.71 |
| 3006 | Entrepeñas | 0.79 | 0.78 | 8.70 | 16.69 | 0.65 | 0.33 | 1.00 | 16.93 |
| 3043 | Buendía | 0.87 | 0.87 | 0.70 | 10.44 | 0.71 | 0.52 | -10.90 | 11.81 |

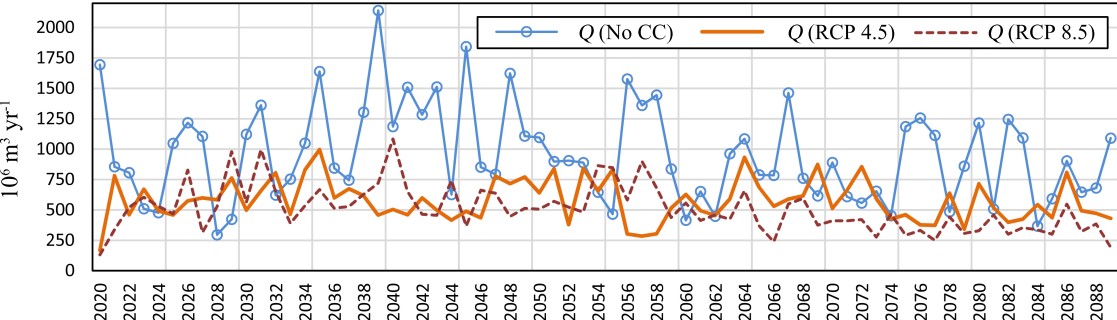

**Figure 7.** Surface flows that enter into the Entrepeñas and Buendía reservoirs (EBR).

decline in resources to $508.9{\cdot}10^6\,\mathrm{m^3\,yr^{-1}}$, on average (Fig. 7). This is due to a combination of a reduction in precipitation (15% and 20% for each scenario, respectively) and an increase in potential evapotranspiration resulting from an increase in temperature of 2.2 °C and 3.4 °C, respectively, for the CC scenarios RCP 4.5 and 8.5.

Regarding the snow-melting modelling, the scenario without CC (historical series) provided an average precipitation as snow ($S_n$) of $185.4{\cdot}10^6\,\mathrm{m^3\,month^{-1}}$ (considering the whole year), reaching a maximum value of $406.9{\cdot}10^6\,\mathrm{m^3\,month^{-1}}$. The hydrological modelling considered the snow for the months between October and May to be relevant, especially that of December, January and February. The snow accumulated on the summits ($S_{np}$) represented an average reserve for each winter of about $112.7{\cdot}10^6\,\mathrm{m^3}$, with a maximum value of $481.5{\cdot}10^6\,\mathrm{m^3}$ (Fig. 8a). For the CC scenarios, the snowfall would

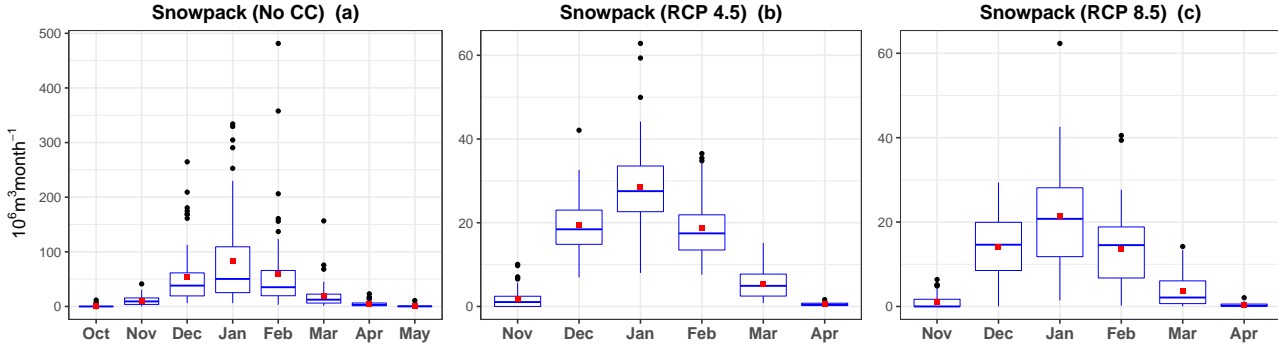

**Figure 8.** Monthly distribution of the snow cover accumulated on the summits ($S_{np}$) in the Tagus Headwaters River Basin (unit: $10^6 \, \mathrm{m^3 \, month^{-1}}$): (a) No CC. (b) RCP 4.5. (c) RCP 8.5.

significantly decrease; in fact, the models did not detect relevant snow-melting processes for the months of October and May. The snowfall would drop by 68% for RCP 4.5, the average value being around $59.3 \cdot 10^6 \, \mathrm{m^3 \, month^{-1}}$, while for RCP 8.5 the snowfall would drop by 90%, giving an average value of about $19.2 \cdot 10^6 \, \mathrm{m^3 \, month^{-1}}$. These values would decrease the snow covers to a similar extent. In the RCP 4.5 scenario the average snow cover of each winter would be $29.5 \cdot 10^6 \, \mathrm{m^3}$ (Fig. 8b),

which represents a reduction of 74%, whereas the RCP 8.5 snow cover would suffer a reduction of 80%, reaching an average value of $22.1 \cdot 10^6 \, \mathrm{m^3}$ (Fig. 8c).

The aquifers recharge estimated by the hydrological modelling ($\Delta$G) had an average value of about $771.0 \cdot 10^6 \, \mathrm{m^3 \, yr^{-1}}$ for the historical climate series, with high monthly variability. This value indicates that almost 80% of the surface flows (Q) have previously passed through aquifers, which underlines the relevance of the groundwater in this river basin. The maximum

recharge occurs during the winter and spring months (January-May), the minimum values being found in the summer months (Fig. 9a). For the CC scenarios, the recharge is significantly reduced and concentrated in the months of February and April. In the RCP 4.5 scenario the aquifers recharge is reduced by 54% (to $357.7 \cdot 10^6 \, \mathrm{m^3 \, yr^{-1}}$) and represents 62% of the surface flow (Q) (Fig. 9b). For RCP 8.5 the aquifers recharge is reduced by 62% (to $290.0 \cdot 10^6 \, \mathrm{m^3 \, yr^{-1}}$) and would represent only 57% of the surface flow (Q) (Fig. 9c). The snowfall reduction is one of the reasons for this decrease in aquifers recharge, since the

melting snow in the model becomes soil moisture (S), as usually happens in nature. These changes in the aquifers recharge alter the pattern of the groundwater discharge ($Q_g$), which is part of the surface flows (Q). In fact, the relative differences in the possible groundwater discharges between the summer and spring months would be greater (Fig. 10a). Overall, the CC scenarios predict a significant reduction in the water resources with respect to the historical climate series, together with a loss of the natural capacity of regulation of the river basin itself, manifested as an increase in the relative values of the maximum

flows (Fig. 10b).

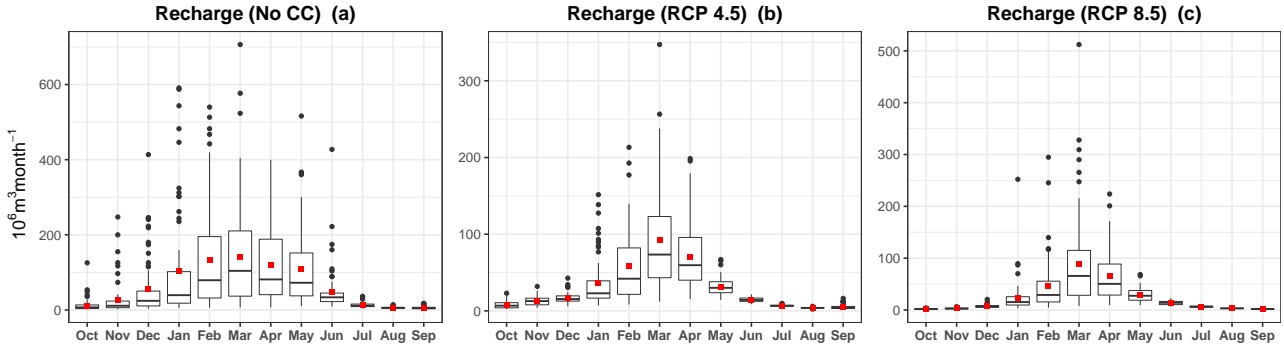

**Figure 9.** Monthly distribution of the recharge ($\Delta$G) in the Tagus Headwaters River Basin (unit: $10^6 \, \mathrm{m}^3 \, \mathrm{month}^{-1}$): (a) No CC. (b) RCP 4.5. (c) RCP 8.5.

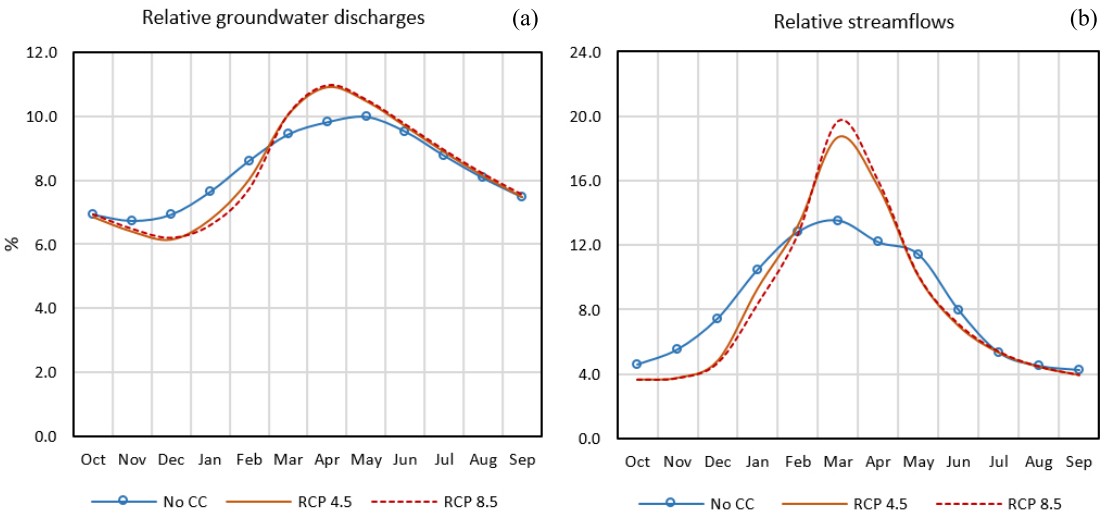

**Figure 10.** Alteration of the intra-annual pattern of the groundwater discharges from aquifers (a) and surface flows (b), in relative terms (%).

## 5.2 Climate change effects on the Tagus Headwaters Water Resources Exploitation System

The simulation of the THWRES was carried out with the flows obtained in the previous hydrological modelling. The results obtained using the historical climate series (No CC) and the stabilisation scenario (RCP 4.5) indicate that the supply to the water uses in the Tagus River Basin is guaranteed, in both cases, for the whole of the time period simulated. However, for the scenario of very-high emissions (RCP 8.5), the simulation indicates that there would be supply deficits in the THWRES in some years towards the end of the time period considered.

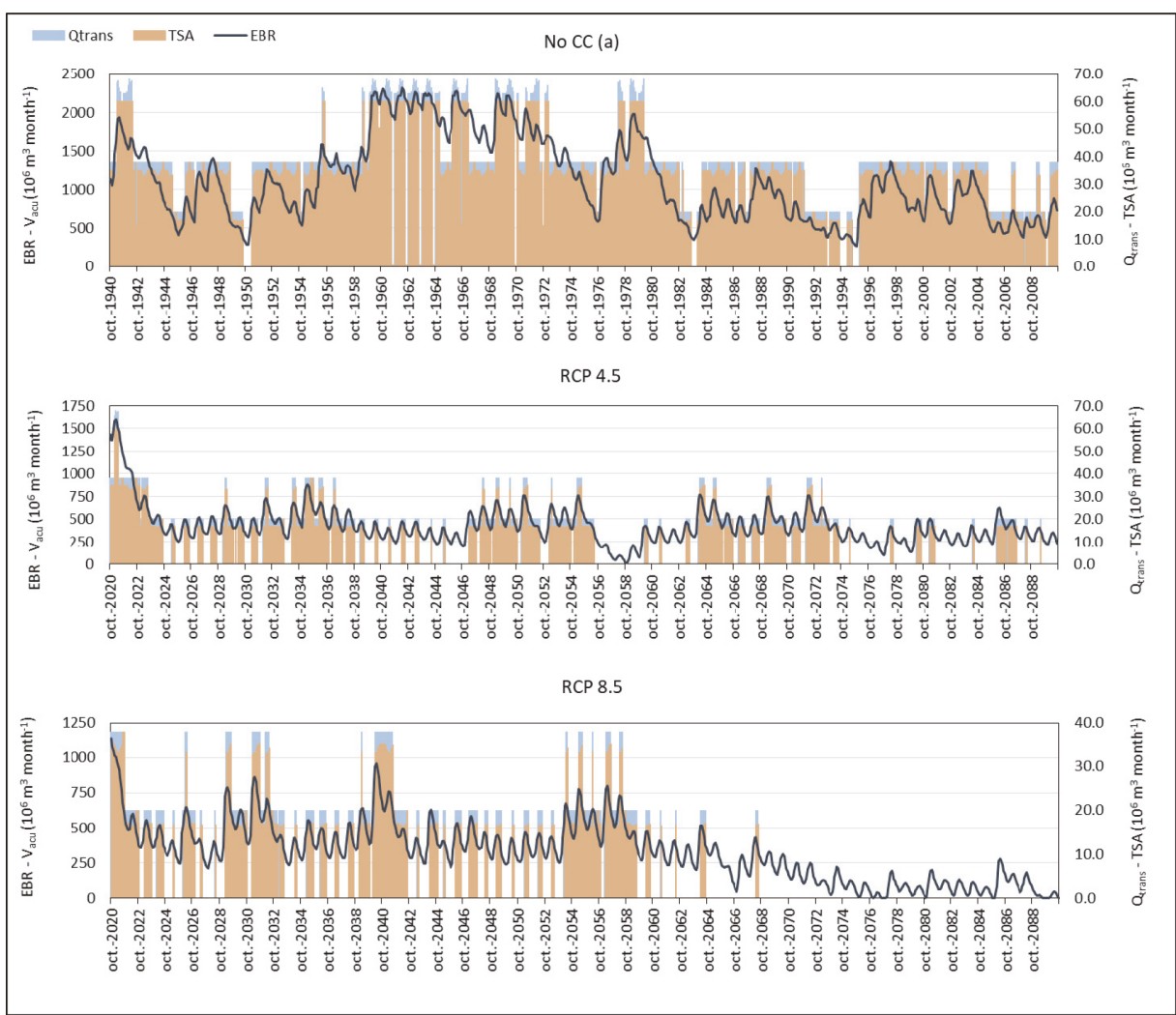

**Figure 11.** The storage volume in the EBR ($V_{acu}$), the total transferred flow ($Q_{trans}$) and the flow to the SRB (TSA) (unit: $10^6 \, \mathrm{m^3 \, month^{-1}}$): a) No CC. b) RCP 4.5. c) RCP 8.5.

Once the water uses in the Tagus River Basin have been supplied, the TSA operating rule comes into effect, providing the monthly volumes that it would be possible to transfer in each climatic scenario evaluated. The results are presented in Figure 11, which shows the volume stored in each month in the EBR ($V_{acu}$), the transferred volume ($Q_{trans}$) that leaves the Tagus River Basin and the flow that would go through the TSA to the SRB.

5    Table 3 summarises the main statistics of the annual transfer series, specifying which part goes to the SRB. If the future climate were similar to that of past years (No CC), the average transferable volume would be about $450 \cdot 10^6 \, \mathrm{m^3 \, yr^{-1}}$ (Fig. 11a). This volume would break down, following the priority criterion established, into about $40 \cdot 10^6 \, \mathrm{m^3 \, yr^{-1}}$ for the Guadiana River Basin ($29 \cdot 10^6 \, \mathrm{m^3 \, yr^{-1}}$ for urban supply and about $10 \cdot 10^6 \, \mathrm{m^3 \, yr^{-1}}$ for Tablas de Daimiel) and $411 \cdot 10^6 \, \mathrm{m^3 \, yr^{-1}}$ for the

**Table 3.** Water volumes transferred ($Q_{trans}$ and TSA) in each climate scenario (unit: $10^6$ m$^3$ yr$^{-1}$).

| | No CC | | RCP 4.5 | | RCP 8.5 | |
|---|---|---|---|---|---|---|
| | $Q_{trans}$ | TSA (to SRB) | $Q_{trans}$ | TSA (to SRB) | $Q_{trans}$ | TSA (to SRB) |
| Average | 449.9 | 410.7 | 142.5 | 123.3 | 99.6 | 86.2 |
| Maximum | 650.0 | 601.4 | 572.6 | 524.2 | 456.1 | 417.3 |
| Minimum | 80.0 | 66.6 | 0.0 | 0.0 | 0.0 | 0.0 |
| Standard deviation | 136.4 | 130.5 | 121.5 | 108.7 | 115.5 | 102.5 |

TSA. Thus, given the average losses by infiltration and evaporation from the transfer channel, which are around 10% (CHS, 2015), the net volume that would reach the SRB would be 370·$10^6$ m$^3$ yr$^{-1}$, a value close to those of the actual series of transfers that have occurred since it came into operation (Morote et al., 2017). So, if there were No CC, despite the fact that the volume reaching the SRB via the TSA would be, on average, much lower than the planned 540·$10^6$ m$^3$ yr$^{-1}$ (600·$10^6$ m$^3$ yr$^{-1}$

minus 10% losses), it would be possible to transfer an average of 411·$10^6$ m$^3$ yr$^{-1}$ to the SRB.

In the RCP 4.5 scenario, the transferable volumes drop to 143·$10^6$ m$^3$ year$^{-1}$ on average, 17·$10^6$ m$^3$ yr$^{-1}$ being for the urban supply in Guadiana River Basin, 2.5·$10^6$ m$^3$ yr$^{-1}$ for Tablas de Daimiel and 123·$10^6$ m$^3$ yr$^{-1}$ for the SRB by means of the TSA infrastructure. So, considering losses of 10%, this would mean net water resources of 111·$10^6$ m$^3$ yr$^{-1}$ being transferred to the SRB, barely 20% of the maximum transferable volume. As worrisome as this important decrease in the average value is

the existence of consecutive periods of three and four years in which no transfer would occur (Fig. 11b).

This situation would be aggravated for the climatic scenario RCP 8.5 since the transferable volume would be reduced to about 100·$10^6$ m$^3$ yr$^{-1}$, on average, distributed as 12·$10^6$ m$^3$ yr$^{-1}$ for the urban supply in Guadiana River Basin, 1.5·$10^6$ m$^3$ yr$^{-1}$ for the maintenance of the Tablas de Daimiel wetland and 86·$10^6$ m$^3$ yr$^{-1}$ for the TSA. Thus, the SRB would receive approximately throughout the year the volume planned for just one month in the operating rule. This scenario worsens the duration

and frequency of the no-transfer periods, which intensify over time, reaching a situation of total cessation of the TSA from the year 2067 due to a lack of accumulated volumes ($V_{acu}$) in the EBR (Fig. 11c).

## 6   Discussion

The hydrological modelling carried out with the AR5 CC scenarios predict a sharp decrease in the water resources of the THRB. These results are in line with the previous simulations made on the same scale (Lobanova et al., 2018), as well as

on a larger scale (Guerreiro et al., 2017; CEDEX, 2011b; Lobanova et al., 2018). The hydrological modelling indicated that the increase in temperature would generate a decline in snowfall, which would leads to a reduction in the snow cover period of two months. The relevant snowfalls would have a delay of one month and the snowmelt would start a month earlier. The decreases provided by the simulations are similar to those published by CEDEX (2017) for the same mountainous area, in which a decrease of up to 70% were predicted for the interval 2070-2100 in the RCP 4.5 scenario, and greater than 90% for the

RCP 8.5 scenario (same interval). In addition, CEDEX (2017) also predicted a decrease in the two-month snow cover period. Regarding the aquifer recharge, the hydrological modelling indicated a decline greater than 50%. Although these values are slightly higher than those published by CEDEX (2017) for the whole Tajo River Basin, they are in line with the previsions in the THRB, since the majority of the projections used for these two scenarios (RCP 4.5 and RCP 8.5) indicated diminishments

of around 50% in aquifer recharge within this area. These alterations of the hydrological processes will probably generate a change in the fluvial regime. In relative terms, the months of autumn (October, November, December) would suffer the major decrease, this change being consistent with the results of Lobanova et al. (2018). At the annual level, these alterations would result in greater variability of the surface flows of the basin due to increases in the relative difference between the maximum and minimum flows of spring and summer, respectively. In short, these changes are going to diminish the natural capacity of

the THRB to regulate its own water resources. However, the results of the THWRES simulations indicate that the EBR will not reach their maximum volume in any situation, and no new infrastructure will be necessary to overcome this loss of regulation.

The effects of these physical changes on the THWRES will translate into average decreases around 70-79% in the volume expected to be available for the TSA in both scenarios evaluated (RCP 4.5 and RCP 8.5). In addition, there would be long periods without transfers for the RCP 4.5 scenario, and this IBWT could even stop operating for the last third of the simulation

period under the RCP 8.5 scenario, when there would be deficits regarding the water uses in the Tagus River Basin itself.

At this point it is necessary to consider two reflections. The first is the uncertainty of regionalised projections, whose origin is associated with the uncertainties inherited during the different stages of their generation. To this uncertainty must be added the influence of uncontrolled local factors in the regionalisation, which have a great relevance in the case of precipitation. In this sense, in Spain, while the regionalised series for temperature are in line with the historical reference series, those of

precipitation have strong uncertainties associated with the models (Amblar-Francés et al., 2017; Mitchell and Hulme, 1999). Even so, the use of these data sources is the best way to understand the repercussions of CC for water resources and hence to adapt strategies to CC. The second is that the simulations carried out for the water resources exploitation system represent scenarios in which the uses of the water and the operating rule do not change with time. The flows that enter in the model are the only input data that vary. So, they are not tools to predict future behaviour, since water needs, infrastructure and water

policy are not static. However, they do serve to evaluate the effects that CC could have on the TSA if measures are not taken regarding the management of the demand for water or the incorporation of alternative sources of water resources into this system.

The socioeconomic impact of the decreases in the TSA supply were calculated by the methodology presented in section 3.3. First, the global decrease in TSA flows to the SRB was distributed among the seven IZs, according to the results obtained

by Martínez-Paz et al. (2016), which were conditioned by the water demand and by the topology of the exploitation system of the SRB. Once these deficits were determined, they were valued using the linear programming presented, calculating GVA and agricultural employment for each IZ according to water allocations available in each scenario. The results are presented in Table 4, which shows the differences in relation to a scenario with no climate change (No CC).

By calculating the ratio between the GVA and TSA the marginal value of water used in irrigation was obtained: 1.29 $€/m^3$

for the RCP 4.5 scenario and 1.31 $€/m^3$ for the RCP 8.5 scenario (averages values for the whole SRB). The marginal value

**Table 4.** Differences of RCP 4.5 and RCP 8.5 scenarios respect to No CC (these values represent decreases with respect to No CC scenario).

| IZ | RCP 4.5 | | | RCP 8.5 | | |
|---|---|---|---|---|---|---|
| | TSA to SRB $(10^6 \, \mathrm{m}^3 \, \mathrm{yr}^{-1})$ | GVA $(10^6 \, € \, \mathrm{yr}^{-1})$ | Labor (full-time jobs $\mathrm{yr}^{-1}$) | TSA to SRB $(10^6 \, \mathrm{m}^3 \, \mathrm{yr}^{-1})$ | GVA $(10^6 \, € \, \mathrm{yr}^{-1})$ | Labor (full-time jobs $\mathrm{yr}^{-1}$) |
| 1 | 54.60 | 79.12 | 1135 | 60.14 | 88.61 | 1251 |
| 2 | 61.60 | 58.33 | 455 | 67.85 | 64.26 | 502 |
| 3 | 22.58 | 34.04 | 651 | 24.87 | 38.65 | 717 |
| 4 | 44.99 | 68.49 | 1484 | 49.55 | 77.26 | 1635 |
| 5 | 93.12 | 122.25 | 2582 | 102.57 | 135.06 | 2845 |
| 6 | 9.41 | 11.51 | 378 | 10.37 | 13.82 | 416 |
| 7 | 8.21 | 5.94 | 3 | 9.04 | 6.79 | 4 |
| SRB | 294.50 | 379.67 | 6690 | 324.40 | 424.46 | 7369 |

establishes the upper limit of the marginal productivity of water, and therefore it represents the maximum payment capacity of the sector for water in each scenario. The figures obtained are in range with those presented in other studies for the area (Martínez-Paz et al., 2016; Albaladejo-García et al., 2018).

Thus, the decrease in the TSA supply for the RCP 4.5 scenario would mean a direct loss of 380 million € $\mathrm{yr}^{-1}$ (at the prices of 2017), while for RCP 8.5 it would amount to 425 million € $\mathrm{yr}^{-1}$. Given that the GVA of irrigation in the SRB is around 1870 million € $\mathrm{yr}^{-1}$ (Martínez-Paz and Pellicer-Martínez, 2018), these figures represent, respectively, direct losses of 20% and 23% in terms of GVA. In addition, the decrease in the volumes transferred in the CC scenarios means that the irrigated area is occupied by more-labour-extensive crops, causing the loss of around 6690 and 7369 direct full-time agrarian jobs per year in the sector (Table 4). To all these direct effects we should add the drag effects of the primary sector on other economic sectors that depend directly on the use of irrigation in the area (food industry, marketing, transport, inputs, etc.) and make up the well-known agro-industrial cluster of the Region of Murcia (Colino et al., 2014). In this sense, some authors propose a multiplier of no less than 3 to calculate the total economic impact of agricultural production on other related activities. To this economic impact it would be possible to add the environmental one (Martínez-Paz et al., 2018), as it would be, for example, the reduction of the flows circulating in the Segura River, since it is part of the distribution network of the TSA volumes, which would also have effects on the ecosystems associated with this river (Perni and Martínez-Paz, 2017).

## 7 Conclusions

In this work, the possible effects of CC on the Tagus Headwaters Water Resources Exploitation System have been evaluated. In particular, the work has focused on the hydrological changes that would arise in the Tagus Headwaters River Basin (THRB) if the predicted climatic scenarios arose, and how these would affect the uses of the basin itself and the IBWT (the TSA) that transfers flows to the Segura River Basin. The main effects on the water cycle would be the reduction of snowfalls and their

accumulation on the summits, and a decrease in aquifers recharge. These changes would generate a significant loss of the natural regulation capacity of the THRB itself, which could not be corrected with new infrastructure since there would also be a significant reduction in the water resources.

The hypothetical flows that can be transferred by the TSA would suffer a significant decrease if the simulated CC scenarios came into being. For RCP 4.5 there would be an average reduction of 70%, while for RCP 8.5 it would be 79%, both figures being with respect to the series without CC. These reductions would result in direct economic losses in the irrigation sector around 20%. Beyond these average figures, the increase in the zero-transfer periods would have a high impact, since it would further increase the uncertainty associated with this source of supply, preventing adequate planning of water uses in the receiving basin.

Finally, in addition to the undoubted interest that presents the case study analysed, this work shows a complete sequential methodological framework which should serve as a guide for the comprehensive evaluation of IBWT within a framework of integrated water resources management. Moreover, the entire methodology has been developed with Open Source tools, facilitating its reproducibility in other areas.

*Author contributions.* Both authors contributed equally to this work.

*Competing interests.* The authors declare that they have no conflict of interest.

*Acknowledgements.* This research work has been supported by project 19342/PI/14 funded by "Fundación Séneca-Agencia de Ciencia y Tecnología de la Región de Murcia" in the framework of PCTIRM 2011-2014. Moreover, authors thank AEMET and UC for the data provided for this work (Spain02v5 dataset, available at http://www.meteo.unican.es/datasets/spain02). We appreciate the valuable comments and suggestions provided by the editor and two anonymous referees.

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
