# Peer review of "Climate change effects on the hydrology of the headwaters of the Tagus River: implications for the management of the Tagus-Segura transfer"

_Hydrology and Earth System Sciences, 2018_

## Referee Comment (RC1) · Anonymous Referee #1 · 2 Jul 2018

GENERAL COMMENTS

The manuscript studies the effects of climate change on the Tagus-Segura water transfer. According to the authors, the work constitutes a new contribution because it analyzes the impact of climate change in an inter-basin water transfer from an integrated water management perspective. Despite they cite a recent article where the issue has already been addressed (Morote et al., 2017), they explain that their approach includes three new aspects: 1) specific modelling of climate scenarios; 2) hydrological modelling and; 3) simulation of the system management under the current operating rule.

[Figure]

In my opinion, despite the subject could be really interesting for the future management of Tagus-Segura water transfer, the selected methodology does not constitute a new contribution to the existing literature. Besides, although the estimation of climate change socioeconomic impacts for the case study could be considered as a novelty, it is scarcely developed. Demand curves are mentioned for the first time in the Discussion section, the methodology to obtain them is not further explained and plots are not provided. Finally, some of the data, methods and assumptions should be deeply reviewed (e.g., the operating rule they apply is no longer into force).

SPECIFIC COMMENTS

Page 3, Line 27: "(...) having a Mediterranean climate". It is more accurate to define the basin climate as Continental-Mediterranean (CHJ, 2015).

Page 4, Lines 7-11: "Moreover, a large part of these water resources (up to 650 106 m3yr−1) are susceptible to being transferred to the neighbouring Guadiana River Basin and, further away, to the SRB. The former can receive up to 50 106m3yr−1, of which 20 106m3 are for the maintenance of the wetland of Tablas de Daimiel and 30 106m3 are for urban use in La Mancha (Albacete), while the SRB can receive up to 600 106m3yr−1 (gross volume), the maximum monthly flow being 60 106m3". According to this paragraph, Guadiana River Basin can receive up to 30 106m3 for urban use in La Mancha (Albacete). Nevertheless, urban supply to the city of Albacete is provided from the Tagus-Segura water transfer (CHJ, 2016). Moreover, the water rights for Albacete and its area of influence are 24,59 millions of cubic meters per year (24 come from surface resources and 0,59 are groundwater) (CHJ, 2016). Authors should clarify this statement and the origin of the data they provide.

Page 4, Line 13 - Page 5, Line 11: "The operating rule (Fig. 2) basically consists of two conditioning factors (BOE, 2014)". The considered operating rule (BOE, 2014) is not longer in force, because the most recent one was included in Law 21/2015.

Page 9, Lines 4-5: "The locations of the 12 gauging stations that have observed flows,

which previously have been naturalized (...)". It is not fully clear if the authors applied a procedure to obtain the naturalized inflows or if these inflows come from the River Basin Authority. In the case they have used the Water Rights Analysis Package (WRAP; Wurbs, 2005), further information should be provided in relation to the demands' supply, reservoir management and other human activities considered to obtain the naturalized inflows. Moreover, it would be interesting to compare the naturalized inflows of the WRAP with the ones of the River Basin Authority. Finally, if WRAP has not been used, the reference should be removed.

Page 9, Lines 7-9: "Each series of observed flows was divided into two periods (Klemeš, 1986), the first (from September 1985 to July 1995) being used in the validation and the second (from August 1995 to December 2009) being used in the calibration". Authors should clarify why they used a calibration period of only 14 years (which I consider is really short for a monthly time step) when there are available inflow time series from 1940 to the current date.

Page 9, Lines 22-24: "Among these series provided by the regionalised models, the 10 that best fitted the historical series for both temperature and precipitation were assembled on a monthly basis (...)" Authors should explain which is the control period considered, as well as the method used to test the goodness of fit and the test results. Which models were finally selected for the basin?

Page 9, Lines 25-27: "Based on the average monthly temperature data of the three climatic scenarios, the potential evapotranspiration series were estimated using the Thornthwaite method (...)". For example, it considers that PET is null when the temperature is near zero. According to the authors, "(...) while the average annual temperature is 11 ◦C, in the coldest months there are values less than zero (...)" (Page 3, Lines 29-30). As Thornthwaite method considers that potential evapotranspiration (PET) is null when the temperature is near zero, it could undervalued the PET in the Tagus basin. Authors should discuss the reasons why this approach was selected.

[Figure]

Page 10, Figure 5: According to the figure, only one of the considered thermo-pluviometric stations (3044) is located inside the basin, which according to the authors has an area of 7.000 km2 (Page 3, Line 26). How could you assure that the temperature is properly characterized, considering that you only take into account 6 points (5 of them outside of the basin), for such a large area? Have you compared the bias between the observed precipitation and temperature values and the modeled ones for the control period of the climate models?

Page 12, Lines 3-5: "The values of the criterion coefficients calculated in the hydrological modelling show that the model employed reproduced properly the surface flows in the THRB: high values of ENS and R2, together with low relative errors (ERMS) and volume errors (PBIAS)". According to Table 2, calibration values are in the range of 0,51-0,87 for the NSE. Nevertheless, NSE values for the validation period oscillate between -0,36 and 0,57. Moreover, 10 out of 12 sub-basins show NSE values below 0,50 during the validation period. Regarding these results, it is not possible to state that the hydrological model is properly reproducing the surface flows during the validation period. Authors should consider model re-calibration or even the use of another hydrological model.

Page 12, Lines 8-11: "The simulation of the historical climate series of 1940-2010 with the calibrated model provided an average annual resource of 954.6 106m3yr−1 (Q). This series was temporally translated to the 2020-2090 time period in order to reproduce a future climate scenario without climate change (No CC). The simulations for climate change scenarios RCP 4.5 and RCP 8.5 with the same calibrated model indicated that the THRB could suffer a considerable loss of its natural water resources". The meaning of "temporally translated" is not clear. Are reduction/increase coefficients obtained from the RCPs applied to the precipitation/temperature 1940-2010 time series to simulate "future conditions"?

Page 12, Lines 12-15: "The RCP 4.5 scenario forecasted a value of 575.6 106m3yr−1, representing a decrease of 39.7 %, while the RCP 8.5 scenario

predicted a 46.6 % decline in resources to 508.9•106m3yr−1, on average (Fig. 7). This is due to a combination of a reduction in precipitation (15% and 20% for each scenario, respectively) and an increase in potential evapotranspiration resulting from an increase in temperature of 2.2°C and 3.4°C, respectively, for the CC scenarios RCP 4.5 and 8.5". These results should be contrasted and discussed in relation to previous studies of climate change impacts under RCP scenarios in the basin (CEDEX, 2017; Guerreiro et al., 2017; Lobanova et al., 2016; Lobanova et al., 2018).

Page 18, Lines 1-5: "The whole irrigated area in the SRB is divided into seven irrigation zones (IZs); for each of them the water demand curve is estimated from a linear programming model that optimises the Gross Value Added (GVA) by the optimal cultivation plans according to the water supply. This crop programming includes the irrigation situations of woody crop maintenance, the change from irrigated to rainfed crops and the abandonment of irrigation polygons, as well as the impact on employment". Demand curves are mentioned for the first time in the Discussion section. If these demand curves are obtained by the authors, further insight should be provided in the Methodology section, and plots included in the Results section. In any other case, sources should be properly referenced.

OTHER COMMENTS

- There is a mispelling in one of the authors' names: it should be Francisco instead of Fancisco - Figure 6. The text does not match the figure (it represents the DSS diagram instead of the distribution of climate stations with AR5 information and the grid obtained from Spain 02v5. - Acknowledgments: As Spain02 v5 datasets have been used, authors should include the following text: "The authors thank AEMET and UC for the data provided for this work (Spain02 v5 dataset, available at http://www.meteo.unican.es/datasets/spain02)" - Page 20. Lines 13-18. There is an updated work from the same institution, which uses the same climate models considered in the present work: CEDEX, 2017. Evaluación del impacto del cambio climático en los recursos hídricos y sequías en España. Informe final. Ministerio de Agricultura,

Alimentación y Medio Ambiente.

- Additional suggested references:

Anastasia Lobanova, Stefan Liersch, Joao Pedro Nunes, Iulii Didovets, Judith Stagl, Shaochun Huang, Hagen Koch, María del Rocío Rivas López, Cathrine Fox Maule, Fred Hattermann, Valentina Krysanova, Hydrological impacts of moderate and high-end climate change across European river basins, Journal of Hydrology: Regional Studies, Volume 18, 2018, Pages 15-30, ISSN 2214-5818, https://doi.org/10.1016/j.ejrh.2018.05.003.

Enze Zhang, Xin'an Yin, Zhihao Xu, Zhifeng Yang. Bottom-up quantification of interbasin water transfer vulnerability to climate change, Ecological Indicators, Volume 92, 2018, Pages 195-206, ISSN 1470-160X, https://doi.org/10.1016/j.ecolind.2017.04.019.

Liz Clarke, Mark Rousevell, Miriam Dunn et al. Climate Change Impacts, Adaptation and Vulnerability Model Applications in Three Regional to Local Scale Case Studies in Europe. IMPRESSIONS Deliverable D3C.2. July 2017. Available online:

Selma B. Guerreiro, Stephen Birkinshaw, Chris Kilsby, Hayley J. Fowler, Elizabeth Lewis. Dry getting drier – The future of transnational river basins in Iberia, Journal of Hydrology: Regional Studies, Volume 12, 2017, Pages 238-252, ISSN 2214-5818, https://doi.org/10.1016/j.ejrh.2017.05.009.

---

## Referee Comment (RC2) · Anonymous Referee #2 · 16 Jul 2018

The paper proposes a methodology for study the effect of climate change and its repercussions on one of the most important water transfers between basins in Europe. To do this, a simple hydrological model (ABCD) is used in the donor basin to analyze the behavior of its components, trying to improve some problems (such as spatial aggregation or incorporate a snow model); after calibrating the model with the algorithm SCE-UA, they simulate and predict the behavior of the system from an ensemble of AR5 models regionalized by the AEMET, observing important reductions in the snowfalls and snow covers, the recharge of aquifers and the available water resources.

[Figure]

a) GENERAL COMMENTS:

I think it's a very relevant topic in semi-arid environments. In these areas, water resources play a very important role, affecting all economic sectors. In this sense, transfers in the Iberian Peninsula have become a factor of territorial disputes over water, and science must have a relevant and objective role, and ultimately, as an impartial arbiter.

Therefore, it is important to carry out this type of study. I believe that the results obtained are very relevant, but I also want to highlight the proposed process. The reproducibility of the process to other basins is very important because it can be used as part of integrated water resources management.

Regarding the scientific evaluation of work, I believe that the paper address relevant scientific questions within the scope of HESS. I believe that the topic is of significance with some degree of originality, and it is relevant to Hydrology and Earth System Sciences. The authors make a clear and concise description of the research problem considered. The introduction and the state of the art were selected correctly, covering current references. They justified the choice of method and the clear connection with the framework. The information sources were collected correctly. As a result, the research objectives are met, obtaining interesting and relevant results for the problem addressed (transfers between basins and their behavior in the future) that can have a significant impact on the topic.

However, the paper presents some aspects that must be reviewed. I will state the improvements that the authors must attend to.

SPECIFIC COMMENTS:

C.1.- First page: Change Fancisco by Francisco.

1. Introduction C.2.- In introduction, the authors introduce the reader to the problem in an appropriate way. However, I believe that the work process is also relevant. After reading the methodology I deduce that they have used Open Source tools. It may be

interesting to include this aspect in the objectives: use of Open Source and free tools to obtain a reproducible process.

Alternatively, they can include this aspect in the conclusions. I believe it is a significant improvement of this work.

C.3.- Figure 1: Do the authors generate the information? If not, they should refer the source in the title.

C.4.- Figure 2: Include the source.

C.5.- Page 6: I think the introduction to this section and Figure 3 are right for a better understanding. However, it should be completed with a paragraph on the implementation of the process: Tools used and interoperability between them, is the process reproducible? They should comment on the programs used (GIS, program for data analysis, etc.)

C.6.- L.8, page 6: I deduce that the authors use ABCD because it is a model that simplifies the process, allowing its understanding with good performance results. I think they should say it in this paragraph, before citing the improvements they make; I agree, it is a model with some problems, such as spatial aggregation.

C.7.- L.3, page 9: The authors use the DEM but they do not explain why. I think they have used it to generate the sub-basins. They must explain it and cite the method used. For example, the D8 algorithm (O'Callaghan et al., 1984).

C.8.- L.11-18, page 9: Why do they use the period 1980-2009? The authors must justify it in the text.

C.9.- L.11-18, page 9: Why use Spain02v5 ?. The authors must clarify it in the text.

C.10.- L.19-29, page 9: The authors have done ensembles based on different models. If so, they must include some agreement metrics of the ensemble.

C.11.- L.19-29, page 9: The authors perform an interpolation of point data to improve

their spatial representation. Although it is not the objective of this work, they should briefly describe this process.

C.12.- L.33, page 9: "... no significant differences ...." Is a statistical test performed to evaluate the significant differences? If not, you should change this phrase to: "...relevant differences..."

C.13.- Figure 6: The caption is incorrect, this figure represents the SIMGES scheme. Change the caption.

C.14.- Table 1: The caption of the table and the title of the first row are the same. Authors can shorten the title to improve the presentation of the table. 5 Results

C.15.- Table 2 and paragraphs related to it: I believe that the result of the adjustment is sufficient at the outlet from the basin to use the model with the RCP scenarios. However, they should comment, when describing the behavior of the components of the model, that a greater uncertainty in some sub-basins should be taken into account (it is observed in the agreement in the validation). I believe that there is greater uncertainty in the headwater basins, probably derived from errors in the validation data (the validation series starts in 1985). This fact can explain the agreement of the calibration against the validation in these basins, while in the output the adjustment is greater. In this period it is probable that the data observed in the output have higher quality compared to the intermediate stations, due to the importance of monitoring at the beginning of the Tajo-Segura transfer. In this period, the data observed are probably more reliable in the output compared to the intermediate stations, due to the importance of monitoring the Tajo-Segura transfer (initiated in the 1980s). However, the reliability of monitoring stations has improved today.

C.16.- Figures 8 and 9: Authors should consider improving the quality of figures. 7 Conclusions

C.17.- L.4-6, page 19: The authors could include in this paragraph the recommendation

in commentary C.2.

References:

C.18.- The authors should review the format of the references, there are some problems. For example, capital letters in the name of some articles. c.19.- Reference Gomariz-Castillo et al. (2015): The authors should change this reference with: Gomariz-Castillo, F., Alonso-Sarría, F. & Cabezas-Calvo-Rubio, F.: Calibration and spatial modelling of daily ET0 in semiarid areas using Hargreaves equation, Earth Science Informatics, 15, 1–16, https://doi.org/10.1007/s12145-017-0327-1. 2017

---

## Author Comment (AC1) · 12 Aug 2018

The paper proposes a methodology for study the effect of climate change and its repercussions on one of the most important water transfers between basins in Europe. To do this, a simple hydrological model (ABCD) is used in the donor basin to analyze the behavior of its components, trying to improve some problems (such as spatial aggregation or incorporate a snow model); after calibrating the model with the algorithm SCE-UA, they simulate and predict the behavior of the system from an ensemble of AR5 models regionalized by the AEMET, observing important reductions in the snowfalls and snow covers, the recharge of aquifers and the available water resources.

Dear Referee,

We would like to thank you for the thorough review and comments. These have helped us to conscientiously review the work and improve it sensibly. We agree with all of them. In addition to the modifications derived from these comments, others have been made motivated by the other reviewer and by errata detected in the review process.

We look forward to any other consideration you consider appropriate.

In the review, the following pattern has been followed:

The replies are in GREEN.

While the new additions to the article are in *ITALICS* and in *BLUE*.

a) GENERAL COMMENTS:

I think it's a very relevant topic in semi-arid environments. In these areas, water resources play a very important role, affecting all economic sectors. In this sense, transfers in the Iberian Peninsula have become a factor of territorial disputes over water, and science must have a relevant and objective role, and ultimately, as an impartial arbiter.

Therefore, it is important to carry out this type of study. I believe that the results obtained are very relevant, but I also want to highlight the proposed process. The reproducibility of the process to other basins is very important because it can be used as part of integrated water resources management.

Regarding the scientific evaluation of work, I believe that the paper address relevant scientific questions within the scope of HESS. I believe that the topic is of significance

with some degree of originality, and it is relevant to Hydrology and Earth System Sciences.

The authors make a clear and concise description of the research problem considered. The introduction and the state of the art were selected correctly, covering current references. They justified the choice of method and the clear connection with the framework. The information sources were collected correctly. As a result, the research objectives are met, obtaining interesting and relevant results for the problem addressed (transfers between basins and their behavior in the future) that can have a significant impact on the topic.

However, the paper presents some aspects that must be reviewed. I will state the improvements that the authors must attend to.

SPECIFIC COMMENTS:

C.1.- First page: Change Fancisco by Francisco.

Thanks for the appreciation. We have corrected the errata.

1. Introduction.

C.2.- In introduction, the authors introduce the reader to the problem in an appropriate way. However, I believe that the work process is also relevant. After reading the methodology I deduce that they have used Open Source tools. It may be interesting to include this aspect in the objectives: use of Open Source and free tools to obtain a reproducible process.

Thanks for the observation and recommendation. We have used Open Source tools to simulate the semi-distributed abcd model. Both data processing and hydrological modelling with R and QGIS. A software that is free for scientific issues (SIMGES from AQUATOOL) has simulated the water exploitation system, so it can also be considered as Open Source tool for researcher community. We have include a reflection about this issue at the end of the introduction:

*"Finally, note that the complete methodology was developed by Open Source tools and by free software for scientific community, which facilities the reproducibility of the work."*

Alternatively, they can include this aspect in the conclusions. I believe it is a significant improvement of this work.

Similarly. In the conclusion, we have include a sentence highlighting this aspect as improvement in this work.

*"Moreover, the complete methodology has been develop with Open Source tools, facilitating its reproducibility in other areas."*

C.3.- Figure 1: Do the authors generate the information? If not, they should refer the source in the title.

Thanks for the question. We have generated the Figure 1 by official source of information. We have referred it in the caption of the Figure 1 *"(CHT, 2018")*.

We have included a new reference:

CHT: Centro de descargas de capas en formato shape de la Cuenca Hidrográfica del Tajo. (In Spanish). Confederación Hidrográfica del Tajo Ministerio de Agricultura, Alimentación y

Medio Ambiente. España. Madrid. http://www.chtajo.es/LaCuenca/Paginas/DescargaDCapas.aspx, 2018.

C.4.- Figure 2: Include the source.

Thanks for the recommendation. We have include the source in the caption of the Figure 2 *"(BOE, 2014; 2015)"*.

BOE: Real Decreto 773/2014, de 12 de septiembre, por el que se aprueban diversas normas reguladoras del trasvase por el acueducto Tajo-Segura. Ministerio de Agricultura, Alimentación y Medio Ambiente. Boletín Oficial del Estado (BOE), 223: 71634-71639. (In Spanish). https://www.boe.es/buscar/doc.php?id=BOE-A-2014-9336, 2014.

BOE: Ley 21/2015, de 20 de julio, por la que se modifica la Ley 43/2003, de 21 de noviembre, de Montes. Jefatura del Estado. Boletín Oficial del Estado (BOE), 173: 60234-600272. (In Spanish). https://www.boe.es/diario_boe/txt.php?id=BOE-A-2015-8146, 2015.

C.5.- Page 6: I think the introduction to this section and Figure 3 are right for a better understanding. However, it should be completed with a paragraph on the implementation of the process: Tools used and interoperability between them, is the process reproducible? They should comment on the programs used (GIS, program for data analysis, etc.)

Thanks for the observation. It is related with the comment 2. We had described the methodology in general way. In this new version of the manuscript, we only name the programs used and their use at the end of the section 3 (Methodology). We also introduce some references. In this point, the DSS SIMGES is named for the first time, which is described later in section "3.2. Simulation of the water resources exploitation system". All the tools used and data is available in internet, so the followed process can be reproducible.

In the section "3. Methodology" we include the following paragraph:

*"The methodological framework stages were developed with Open Source tools. QGIS (QGIS Development Team, 2016) were use in the data processing of spatial information, R was employed for data analysis and hydrological modeling (R Core Team, 2016), whereas the DSS SIMGES was used for the simulation of the water resources exploitation system (Pedro-Monzonis et al, 2016a)."*

We have included these references:

QGIS Development Team.: QGIS Geographic Information System. Retrieved from http://qgis.osgeo.org., 2016.

R Core Team.: R: A Language and Environment for Statistical Computing. Vienna, Austria. Retrieved from https://www.r-project.org/., 2016.

C.6.- L.8, page 6: I deduce that the authors use ABCD because it is a model that simplifies the process, allowing its understanding with good performance results. I think they should say it in this paragraph, before citing the improvements they make; I agree, it is a model with some problems, such as spatial aggregation.

Thanks for the recommendation. We have tested some lumped water balance model in advance: WAPABA (Wang et al., 2011), WASMOD (Kizza et al., 2011), GR2 (Makhlouf

and Michel, 1994), and Thortwaite-Mather (Alley et al., 1984), and the abcd model was the model which provides the best performance ($E_{NS}$). As it is a lumped water balance that simplifies the water cycle into two storages using only four parameters, it is possible understand how the model operates.

In addition, we have applied in semi-distributed manner in order to improve the possible deficiencies by spatial aggregation. Therefore, the simple structure of the model allows understanding the followed methodology, which guarantee its reproducibility in other areas.

We have included a reflection about the reproducibility at the end of this paragraph. We consider that it is not appropriate to indicate the performance of the model in advance. The new sentence is:

*"This water balance model and this structure were selected in order to facilitate the understanding of the developed process, allowing the possible reproducibility of the work."*

These references are not included in the manuscript:

Wang, Q. J.,et al., 2011. Monthly versus daily water balance models in simulating monthly runoff. Journal of Hydrology, 404(3-4), 166-175. doi: 10.1016/j.jhydrol.2011.04.027

Makhlouf, Z. and Michel, C. 1994. A 2-parameter monthly Water-Balance Model for French watersheds. Journal of Hydrology, 162(3-4), 299-318. doi: 10.1016/0022-1694(94)90233-X

Kizza, M., *et al.*, 2011. Modelling catchment inflows into Lake Victoria: uncertainties in rainfall-runoff modelling for the Nzoia River. *Hydrological Sciences Journal-Journal des Sciences Hydrolgiques*, 56(7), 1210-1226. doi: 10.1080/02626667.2011.610323

Alley, W.M., 1984. On the Treatment of Evapotranspiration, soil-moisture accounting, and aquifer recharge in monthly water-balance models. Water Resources Research, 20(8), 1137-1149. doi: 10.1029/WR020i008p01137

C.7.- L.3, page 9: The authors use the DEM but they do not explain why. I think they have used it to generate the sub-basins. They must explain it and cite the method used. For example, the D8 algorithm (O'Callaghan et al., 1984).

Thanks for the feedback and the reference. We have used the DEM to delimitate the catchments and to process the climatic variables. For the case of catchments delimitation, we have used the algorithm D8 since it is already implemented in QGIS software. So, we have include this reference in the text.

We have made a modification in the manuscript at the beginning of the section "4.1. Hydrological modelling of the Tagus Headwaters River Basin".

*"The Digital Elevation Model (DEM) employed has a 25-m resolution and is available on the website of the National Geographic Institute of Spain (www.cnig.es). This DEM was used as an auxiliary variable in the interpolation models of the climatic variables, and to delimit the main streams and catchments using D8 algorithm (O'Callaghan et al., 1984). The locations of the 12 gauging stations, which have observed flows in the same period, were used to establish the outlet points of the 12 catchments in which the THRB was divided (Fig. 5)."*

This reference has been included:

> O'Callaghan, J. F., Mark, D. M.: The extraction of drainage networks from digital elevation data. Computer Vision, Graphics and Image Processing, 28(1), 328–344, 1984.

C.8.- L.11-18, page 9: Why do they use the period 1980-2009? The authors must justify it in the text.

Thanks for the recommendation. We have used this period since there are data for all the gauging stations at the same time. We introduce two sentences clarifying this question in the section "4.1. Hydrological modelling of the Tagus Headwaters River Basin".

*"… The locations of the 12 gauging stations, which have observed flows in the same period, were*

*used to establish the outlet points of the 12 catchments in which the THRB was divided (Fig. 5)…*

*As there are data available to the 12 gauging stations for the same period, it is possible to calibrate*

*the model jointly for all the catchments."*

To complete this statement, we present the following reflection:

In the study area, there are the following data of gauging stations and measurements in the reservoirs:

[Figure]

[Figure]

[Figure]

So, to calibrate the abcd model in a semi-distributed way, in order to capture the heterogeneity of the Headwaters of the Tagus River Basin (THRB), we used the period 1985-2010 so that there are data from the 12 gauging stations at the same period.

C.9.- L.11-18, page 9: Why use Spain02v5 ?. The authors must clarify it in the text.

Thanks for the question. The advantages of using Spain02v5 are that their daily series of precipitation and temperature are refined (without outliers and/or inhomogeneities), and that they include the spatial variability of the climatic variables in a grid with a 12.5-km resolution. Thus, as in Tagus Headwater River Basin there are areas without climatic measurements, the spatial variability associated with the climatic variability can be included using this data source, as it is said in Herrera et al. (2016).

We have included this clarification in the section "4.1. Hydrological modelling of the Tagus Headwaters River Basin".

*"…The advantages of using Spain02v5 are that the daily series of precipitation and temperature are refined (without outliers and/or inhomogeneities), and that they include the spatial variability of the climatic variables in a grid with a 12.5-km resolution (**Fig. 5**)."*

C.10.- L.19-29, page 9: The authors have done ensembles based on different models. If so, they must include some agreement metrics of the ensemble.

Thanks for the observation. In the manuscript, we have specified the models used to make the ensembles: Simple Average Forecast Combination (SA) and Bias-Corrected Eigenvector Combination (EIG2). In addition, the main results of the goodness of fit in

the adjustments are included ($E_{NS}$). For example, the $E_{NS}$ values obtained with EIG2 for the temperatures were 0.87, while in the separate models it never exceeded 0.77. For the precipitations, the $E_{NS}$ value was 0.30, while in the models it was always lower than 0.

We have included the name of the ensembles and the metrics in the section "4.1. Hydrological modelling of the Tagus Headwaters River Basin".

*"…The 10 models that best fitted for both temperature and precipitation were assembled using the Simple Average Forecast Combination (SA), and Bias-Corrected Eigenvector Forecast Combination (EIG2) (Hsiao and Wan, 2014), available in the R-CRAN package GeomComb (Weiss and Roetzer, 2016). Then, both ensembles have also been compared with Spain02v5 data using the same control period (1971-2005). Finally, the series obtained by EIG2 have been used, with a lower prediction error compared to the rest of series. For example, the $E_{NS}$ values obtained with EIG2 for the temperatures were 0.87, while in the separate models it never exceeded 0.77. For the precipitations, the $E_{NS}$ value was 0.30, while in the models it was always lower than 0. In addition, another advantage of EIG2 method is that it allows to correct the bias produced by predictive models. ..."*

C.11.- L.19-29, page 9: The authors perform an interpolation of point data to improve their spatial representation. Although it is not the objective of this work, they should briefly describe this process.

Thanks for the suggestion. The algorithm used to carry out the interpolations of the climatic variables is included in the manuscript (Thin-Plate Splines method), at the end of the section "4.1. Hydrological modelling of the Tagus Headwaters River Basin".

*"Once the monthly series of precipitation, temperature and potential evapotranspiration had been calculated, they were spatially interpolated on the cells using the Thin-Plate Splines (TPS) method (Wahba, 1990), which is based on local interpolation from polynomials. This method turns out to be relatively robust against non-compliance with the statistical assumptions necessary in methods such as Kriging, being used with good results for the interpolation of climatic variables such as rainfall (Hutchinson, 1995) and temperatures (McKenney and others, 2006). Finally, each basin was assigned the average value of the cells over which it extends."*

C.12.- L.33, page 9: "... no significant differences ...." Is a statistical test performed to evaluate the significant differences? If not, you should change this phrase to: "...relevant differences..."

Thanks for the correction. We have changed "…no significant differences…" by *"…relevant differences…".*

C.13.- Figure 6: The caption is incorrect, this figure represents the SIMGES scheme. Change the caption.

Thanks for the observation. It has been a misprint. We have corrected the title of this Figure that coincided with Figure 5.

The correct caption is:

*"Fig. 6. Scheme of the water resources exploitation system of the Tagus Headwaters, as far as Aranjuez"*

C.14.- Table 1: The caption of the table and the title of the first row are the same. Authors can shorten the title to improve the presentation of the table.

Thanks for the observation. We have shortened the first row in the Table 1 since the information is duplicated.

Now the first row is:

*"Water uses in THWRES"*

5 Results

C.15.- Table 2 and paragraphs related to it: I believe that the result of the adjustment is sufficient at the outlet from the basin to use the model with the RCP scenarios. However, they should comment, when describing the behavior of the components of the model, that a greater uncertainty in some sub-basins should be taken into account (it is observed in the agreement in the validation). I believe that there is greater uncertainty in the headwater basins, probably derived from errors in the validation data (the validation series starts in 1985). This fact can explain the agreement of the calibration against the validation in these basins, while in the output the adjustment is greater.

In this period it is probable that the data observed in the output have higher quality compared to the intermediate stations, due to the importance of monitoring at the beginning of the Tajo-Segura transfer. In this period, the data observed are probably more reliable in the output compared to the intermediate stations, due to the importance of monitoring the Tajo-Segura transfer (initiated in the 1980s). However, the reliability of monitoring stations has improved today.

Thanks for this interesting reflection. The referee #1 also refers to this question. The uncertainty of the data of observed flows could explain a low adjustment in some upper catchments, as you point out. Although the data series used come from official sources (yearbook of Official Gauging Station Network of Spain) and they have been purified before publication. If there would be errors in the measurements, they would be more probable in the older data since the gauging stations could be not properly calibrated. It could explain the results of the simulation of the Buendía reservoirs inflows by Lobanova et al. (2016). They obtained a worse goodness of fit in the calibration (NSE = 0.39 for the period 1987-1993) than the validation period (NSE = 0.76 for the period 1994-1999). In addition to this, if there would be errors, they can have more influence on the results in the gauging stations of the upper catchments since they have lower flows.

To this, we can add the fact of using the validation period just after the warming up period (the last being the calibration period). Thereby, it can cause that, in some catchments, the warming up period can extend to a part of the validation period. This would entail that the calibration process used the warming up period and a part of the validation to adjust the initial parameters, obtaining worse adjustments in the validation of the model. The union of both sources of uncertainty may be the reasons why the validation yields low values in the goodness of fit. However, as the ultimate goal is to use a calibrated model for the last few years, so the result obtained in the performance is considered as good.

The first paragraph in the section "5.1. Climate change effects on the hydrology of the Tagus Headwaters River Basin" is:

*"The values of the criterion coefficients calculated in the hydrological modelling show that the model employed reproduced properly the surface flows in the THRB in the calibration period: high values of $E_{NS}$ and $R^2$, together with low relative errors ($E_{RMS}$) and volume errors ($P_{BIAS}$). However, in the validation period, there are some low values for the goodness of fit coefficients calculated, indicating that the results of these catchments have greater uncertainty. These results can be explained by the fact that of using the validation period just after the warming up. Thereby, it could cause that, in some catchments, the warming up can extend to a part of the validation period. This would entail that the calibration process used a part of the validation to adjust the initial*

*parameters, obtaining worse adjustments in the validation of the model. But, it is important to highlight that the parameters used in the simulations are adjusted with the more recent data, providing a good performance of the surface flows in the THRB. In addition,* the best performance in calibration corresponded to the outlets of the catchments of Entrepeñas and Buendía, which were the flows used as the input in the subsequent simulation of the water resources exploitation system. In fact, both catchments had $N_{SE}$ values around 0.80 and low $P_{BIAS}$ (**Table 2**)."

C.16.- Figures 8 and 9: Authors should consider improving the quality of figures. 7

Thanks for the observation. We have improved the quality of Figure 7.

Conclusions

C.17.- L.4-6, page 19: The authors could include in this paragraph the recommendation in commentary C.2.

Thanks for the recommendation. We have included this reflection in the section "7. Conclusion".

*"Moreover, the complete methodology has been develop with Open Source tools, facilitating its reproducibility in other areas."*

References:

C.18.- The authors should review the format of the references, there are some problems.

For example, capital letters in the name of some articles.

Thanks for the recommendation. We have reviewed the references.

c.19.- Reference Gomariz-Castillo et al. (2015): The authors should change this reference by:

Gomariz-Castillo, F., Alonso-Sarría, F. & Cabezas-Calvo-Rubio, F.: Calibration and spatial modelling of daily ET0 in semiarid areas using Hargreaves equation, Earth Science Informatics, 15, 1–16, https://doi.org/10.1007/s12145-017-0327-1. 2017

Thanks for the observation. We have updated this reference.

---

## Author Comment (AC2) · 12 Aug 2018

Dear Referee,

We would like to thank you for the thorough review and comments. These have helped us to conscientiously review the work and improve it sensibly. We agree with all of them. In addition to the modifications derived from these comments, others have been made motivated by the other reviewer and by errata detected in the review process.

We look forward to any other consideration you consider appropriate.

The comments replies and proposed modifications can be found in the following at-
tached document.

Sincerely yours

"In the review, the following pattern has been followed: The replies are in GREEN.
While the new additions to the article are in ITALICS and in BLUE."

Please also note the supplement to this comment:
https://www.hydrol-earth-syst-sci-discuss.net/hess-2018-258/hess-2018-258-AC2-
supplement.pdf
258, 2018.

[Figure]

**Supplement:**

GENERAL COMMENTS

The manuscript studies the effects of climate change on the Tagus-Segura water transfer. According to the authors, the work constitutes a new contribution because it analyzes the impact of climate change in an inter-basin water transfer from an integrated water management perspective. Despite they cite a recent article where the issue has already been addressed (Morote et al., 2017), they explain that their approach includes three new aspects: 1) specific modelling of climate scenarios; 2) hydrological modelling and; 3) simulation of the system management under the current operating rule.

In my opinion, despite the subject could be really interesting for the future management of Tagus-Segura water transfer, the selected methodology does not constitute a new contribution to the existing literature. Besides, although the estimation of climate change socioeconomic impacts for the case study could be considered as a novelty, it is scarcely developed. Demand curves are mentioned for the first time in the Discussion section, the methodology to obtain them is not further explained and plots are not provided. Finally, some of the data, methods and assumptions should be deeply reviewed (e.g., the operating rule they apply is no longer into force).

Dear Referee,

We would like to thank the referee for the thorough review and comments, which have helped us to conscientiously review the work and improve it sensibly. We agree with all of them. Moreover, besides the modifications that derive from these comments, others must be made have been made motivated by the other reviewer and by errata detected in the review process.

We look forward to any other consideration you consider appropriate.

In the review, the following pattern has been followed:

The replies are in GREEN.

While the new additions to the article are in *ITALICS* and in *BLUE*.

SPECIFIC COMMENTS

Page 3, Line 27: "(...) having a Mediterranean climate". It is more accurate to define the basin climate as Continental-Mediterranean (CHJ, 2015).

We agree and appreciate this improvement. This definition has been included in the work.

*"…having a Continental Mediterranean (CHJ, 2015) climate…"*

We have included this reference:

CHJ: Plan Hidrológico de la Demarcación Hidrográfica del Júcar. Memoria. Ciclo de Planificación Hidrológica 2015-2021. Confederación Hidrográfica del Júcar. Ministerio de Agricultura, Alimentación y Medio Ambiente. https://www.chj.es/es-es/medioambiente/planificacionhidrologica/Paginas/PHC-2015-2021-Plan-Hidrologico-cuenca.aspx, 2016.

Page 4, Lines 7-11: "Moreover, a large part of these water resources (up to 50ăA ́c106 m3yr☐1) are susceptible to being transferred to the neighbouring Guadiana River Basin and, further away, to the SRB. The former can receive up to 50ăA ́c106m3yr☐1, of which 20ăA ́c106m3 are for the maintenance of the wetland of Tablas de Daimiel and 30ăA ́c106m3 are for urban use in La Mancha (Albacete), while the SRB can receive up to 600ăA ́c106m3yr☐1 (gross volume), the maximum monthly flow being 60ăA ́c106m3". According to this paragraph, Guadiana River Basin can receive up to 30ăA ́c106m3 for urban use in La Mancha (Albacete). Nevertheless, urban supply to the city of Albacete is provided from the Tagus-Segura water transfer (CHJ, 2016). Moreover, the water rights for Albacete and its area of influence are 24,59 millions of cubic meters per year (24 come from surface resources and 0,59 are groundwater) (CHJ, 2016). Authors should clarify this statement and the origin of the data they provide.

Thanks for the correction since including the words "La Mancha (Albacete)" leads to error. A part of "La Mancha (Albacete)" belongs to the Guadiana River Basin (CHG), and another part is within the Jucar River Basin (CHJ).

The volume of 30 $10^6$ m$^3$/year comes from the Transfer Law (BOE, 2014; 2015) and the Guadiana River Basin Plan (CHG, 2016). Whereas the water uses of the city of Albacete and its surroundings belong to the Jucar River Basin (CHJ, 2016) and this volume does not come from the Tagus Transfer. This is the reason because these values do not match.

Therefore, in order to correct this error, we have changed the text in the manuscript to be more accurate. The words "La Mancha (Albacete)" have been eliminated, and they have been changed by "…urban supply to the populations located at the upper of the Guadiana River Basin". Besides, the whole manuscript has been reviewed in order to change these words, for example in the Table 1 the words "La Mancha (Albacete)" have been removed. And in the Figure 6 the name of the water use has also been changed by "Urban Suppy (Guadiana)". In the results section (5.2), we have also changed "La Mancha" by "urban supply in Guadiana River Basin".

We have included the reference to the Basin Plan of the Guadiana River in which these data are established (CHG, 2016).

CHG: Plan Hidrológico de la Parte Española de la Demarcación Hidrográfica del Guadiana. Memoria (Parte I). (In Spanish). Confederación Hidrográfica del Guadiana. Ministerio de Agricultura, Alimentación y Medio Ambiente. http://planhidrologico2015.chguadiana.es/?corp=planhidrologico2015&url=61, 2016.

Page 4, Line 13 - Page 5, Line 11: "The operating rule (Fig. 2) basically consists of two conditioning factors (BOE, 2014)". The considered operating rule (BOE, 2014) is no longer in force, because the most recent one was included in Law 21/2015.

Thanks for this key contribution. We have included the new data in the model that simulates the Tagus Headwaters Water Resources Exploitation System (THWRES). The changes are:

- The accumulated volume in the reservoirs Vacu, that previously was 1300 $10^6$ m$^3$ in the BOE (2014) has been changed by 1500 $10^6$ m$^3$ (BOE, 2015). This affects to the levels 1 and 2.
- The flows that enter in the reservoirs EBR (Aacu) during a year decrease to 1000 $10^6$ m$^3$ (before was 1200 $10^6$ m$^3$ in BOE (2014)). This affects to the levels 1 and 2.

- In addition, the maximum volume that can be transferred in one month at the level 1 increases to 68 $10^6$ m$^3$ (previously 60 $10^6$ m$^3$ in BOE (2014)).

The rest of the conditions remain the same.

Therefore, on the one hand the operating rule is more restrictive, increasing the accumulated volume in the reservoirs for the maximum flow transferred in one month. But, on the other hand the operating rule is less restrictive diminishing the condition of the flows that enter in the reservoirs, and increasing the maximum volume that can be transferred in one month (level 1).

So, the three scenarios have been modeled again with the updated operating rule and the transferred volumes increase slightly in all of them. This leads to slight changes in the figures of the part of results "section 5.2" and in the part of the socioeconomic study of the "6. Discussion". The economic figures of the abstract are also changed. Now the economic losses are between "*380-425 million €*" (in the previous version are between 330-380 million €).

We have included this new reference:"

BOE: Ley 21/2015, de 20 de julio, por la que se modifica la Ley 43/2003, de 21 de noviembre, de Montes. Jefatura del Estado. Boletín Oficial del Estado (BOE), 173: 60234-600272. (In Spanish). https://www.boe.es/diario_boe/txt.php?id=BOE-A-2015-8146, 2015.

The main changes in the manuscript are:

"2. The water resources exploitation system of the Tagus Headwaters River Basin"

*The four levels are:*

- *Level 4. When $V_{acu}$ is lower than $400 \cdot 10^6$ m$^3$. Transfers are not allowed ($Q_{trans}$ = 0 m$^3$/month).*

- *Level 3. When $V_{acu}$ is between $400 \cdot 10^6$ m$^3$ and the values indicated in **Fig. 2**, which vary between $586 \cdot 10^6$ m$^3$ and $688 \cdot 10^6$ m$^3$ depending on the month. Transfers ($Q_{trans}$) of $20 \cdot 10^6$ m$^3$/month are allowed.*

- *Level 2. When $V_{acu}$ is between the volumes established in Level 3 and $1500 \cdot 10^6$ m$^3$, and in addition $A_{acu}$ is lower than $1000 \cdot 10^6$ m$^3$. Transfers ($Q_{trans}$) of $38 \cdot 10^6$ m$^3$/month are allowed.*

- *Level 1. When $V_{acu}$ is equal to or greater than $1500 \cdot 10^6$ m$^3$, or $A_{acu}$ is equal to or greater than $1000 \cdot 10^6$ m$^3$. Transfers ($Q_{trans}$) of $68 \cdot 10^6$ m$^3$/month are allowed.*

[Figure]

**Fig. 2**. *Operating rule of the TSA. Based on BOE (2014; 2015).*

"5.2. Climate change effects on the Tagus Headwaters Water Resources Exploitation System"

**Fig. 11.** The storage volume in the EBR ($V_{acu}$), the total transferred flow ($Q_{trans}$) and the flow to the SRB (TSA) (unit: $10^6 \cdot m^3$/month): a) No CC. b) RCP 4.5. c) RCP 8.5.

[Figure]

**Table 3**. *Water volumes transferred ($Q_{trans}$ and TSA) in each climate scenario (unit: $10^6 \cdot m^3$/year).*

|  | No CC | | RCP 4.5 | | RCP 8.5 | |
|---|---|---|---|---|---|---|
|  | $Q_{trans}$ | TSA (to SRB) | $Q_{trans}$ | TSA (to SRB) | $Q_{trans}$ | TSA (to SRB) |
| Average | 449.9 | 410.7 | 142.5 | 123.3 | 99.6 | 86.2 |
| Maximum | 650 | 601.4 | 572.6 | 524.2 | 456.12 | 417.3 |
| Minimum | 80 | 66.6 | 0.0 | 0.0 | 0.0 | 0.0 |
| Standard deviation | 136.4 | 130.5 | 121.5 | 108.7 | 115.5 | 102.5 |

*If the future climate were similar to that of past years (No CC), the average transferable volume would be about $450 \cdot 10^6$ m³/year (**Fig. 11a**). This volume would break down, following the priority criterion established, into about $40 \cdot 10^6$ m³/year for the Guadiana River Basin ($29 \cdot 10^6$ m³/year for urban supply and about $10 \cdot 10^6$ m³/year for Tablas de Daimiel) and $411 \cdot 10^6$ m³/year for the TSA. Thus, given the average losses by infiltration and evaporation from the transfer channel, which are around 10% (**CHS, 2016**), the net volume that would reach the SRB would be $370 \cdot 10^6$ m³/year, a value close to those of the actual series*

*of transfers that have occurred since it came into operation (**Morote et al. al., 2017**). So, if there were no CC, despite the fact that the volume reaching the SRB via the TSA would be, on average, much lower than the planned 540·10⁶ m³/year (600·10⁶ m³/year minus 10% losses), it would be possible to transfer an average of 411·10⁶ m³/year to the SRB.*

*In the RCP 4.5 scenario, the transferable volumes drop to 143·10⁶ m³/year on average, 17·10⁶ m³/year being for the urban supply in Guadiana River Basin, 2.5·10⁶ m³/year for Tablas de Daimiel and 123·10⁶ m³/year for the SRB by means of the TSA infrastructure. So, considering losses of 10%, this would mean net water resources of 111·10⁶ m³/year being transferred to the SRB, barely 20% of the maximum transferable volume. As worrisome as this important decrease in the average value is the existence of consecutive periods of three and four years in which no transfer would occur (**Fig. 11b**).*

*This situation would be aggravated for the climatic scenario RCP 8.5 since the transferable volume would be reduced to about 100·10⁶ m³/year, on average, distributed as 12·10⁶ m³/year for the urban supply in Guadiana River Basin, 1.5·10⁶ m³/year for the maintenance of the Tablas de Daimiel wetland and 86·10⁶ m³/year for the TSA. Thus, the SRB would approximately receive throughout the year the volume planned for just one month in the operating rule. This scenario worsens the duration and frequency of the no-transfer periods, which intensify over time, reaching a situation of total cessation of the TSA from the year 2067 due to a lack of accumulated volumes ($V_{acu}$) in the EBR (**Fig. 11c**).*

Page 9, Lines 4-5: "The locations of the 12 gauging stations that have observed flows, which previously have been naturalized (...)". It is not fully clear if the authors applied a procedure to obtain the naturalized inflows or if these inflows come from the River Basin Authority. In the case they have used the Water Rights Analysis Package (WRAP; Wurbs, 2005), further information should be provided in relation to the demands' supply, reservoir management and other human activities considered to obtain the naturalized inflows. Moreover, it would be interesting to compare the naturalized inflows of the WRAP with the ones of the River Basin Authority. Finally, if WRAP has not been used, the reference should be removed.

In this work we do not use the WRAP program. It has been a mistake when entering the reference with the references manager. We wanted to use a reference from the same author (R.A. Wurbs) about naturalization of the streamflows (2006). Although we do not use all the techniques explained in this work, it has served as a reference to make our own naturalization of the measurements in the gauging stations. This author also designed the WAM model for the naturalization the streamflows. Although the use of this program (WAM) could have been possible, as other authors do in the interesting recommended reference (Lobanova et al., 2018), we have not considered it appropriate to apply it given the small size of the study area.

In the naturalization of the flows at gaged sites, we have taken into account the regulation and evaporation made by the main reservoirs in the study area (data from the Official Gauging Station Network of Spain (ROEA: http://ceh-flumen64.cedex.es/anuarioaforos/default.asp) and technical reports of the water board: CHT). We have also considered the irrigation and urban supplies during the study period, and the water volumes that the Trillo nuclear power plant has consumed in its period of operation. The flow returns of the water uses have been also considered.

We have not compared the naturalized flow series with the ones of the River Basin Authority. The current official series come from the SIMPA hydrological model, and the previous official

series come from the Sacramento hydrological model. Therefore, we do not consider it appropriate to compare a series of naturalized flow with the results of a model.

We include sentences indicating how we have made the naturalization of the observed flows in the section "4.1. Hydrological modelling of the Tagus Headwaters River Basin".

*"…These observed flows have been previously naturalized to be used in the calibration-validation process (Wurbs, 2005). For that, the main human alterations located upstream of each gauging station were undone: regulation and evaporation in the reservoirs, as well as the derivations for urban, agricultural and industrial uses (also the returns of these uses were considered)…"*

The following reference of Wurbs R.A. has been removed.

Wurbs, R.A. 2005. Modeling river/reservoir system management, water allocation, and supply reliability. Journal of Hydrology, 300(1-4), 100-113.

And we include the correct reference of Wurbs R.A.:

Wurbs, R.A.: Methods for developing naturalized monthly flows at gaged and ungagged sites, J. Hydrol. Eng., 11(1), 55-64, https://doi.org/10.1061/(ASCE)1084-0699(2006)11:1(55), 2006.

Page 9, Lines 7-9: "Each series of observed flows was divided into two periods (Klemeš, 1986), the first (from September 1985 to July 1995) being used in the validation and the second (from August 1995 to December 2009) being used in the calibration". Authors should clarify why they used a calibration period of only 14 years (which I consider is really short for a monthly time step) when there are available inflow time series from 1940 to the current date.

Thanks for this reflection. In the study area, there are the following data of gauging stations and measurements in the reservoirs:

[Figure]

[Figure]

[Figure]

So, to calibrate the abcd model in a semi-distributed way, in order to capture the heterogeneity of the Headwaters of the Tagus River Basin (THRB), we used the period 1985-2010 so that there are data from the 12 gauging stations in the same period. Due to this length, we have chosen to split the period into two periods: 2/3 for calibration and 1/3 for validation (strategy to obtain a significant calibration and data for its testing, as Klemes says), and as is usually done in calibration-validation in hydrology modelling.

For instance, the following works use this kind of partition in the data series for calibration-validation: DAGGUPATI et al. (2015), SHAWUL et al. (2013) or TEGEGNE et al. (2017).

Daggupati, P., Yen, H., White, M.J., Srinivasan, R., Arnold, J.G., Keitzer, C.S., Sowa, S.P. (2015): "Impact of model development, calibration and validation decisions on hydrological simulations in West Lake Erie Basin". Hydrological Process, vol. 29, p. 5307–5320.

Gyamfi, C., Ndambuki, J.M., Salim, R.W. (2016): "Application of SWAT Model to the Olifants Basin: Calibration, Validation and Uncertainty Analysis". Journal of Water Resource and Protection, vol. 8, ID: 65180.

Shawul, A.A., Alamirew, T. Y Dinka, M.O. (2013): "Calibration and validation of SWAT model and estimation of water balance components of Shaya mountainous watershed, Southeastern Ethiopia". Hydrological and Earth System Sciences, vol. 10, p. 13955–13978.

Tegegne, G., Park, D.K, Kim, Y.-O. (2017): "Comparison of hydrological models for the assessment of water resources in a data-scarce region, the Upper Blue Nile River Basin". Journal of Hydrology: Regional Studies, vol. 14, p. 49–66.

In addition, the length of data used is greater than that of the study conducted on the Tagus River in the recommended references of Lobanova et al. (2016; 2018). In both studies, authors use the period 1987-1993 for calibration and 1994-1999 for validation, using the month as time step in the calibration.

We introduce two sentences clarifying this question in the section "4.1. Hydrological modelling of the Tagus Headwaters River Basin".

*"… The locations of the 12 gauging stations, which have observed flows in the same period, were used to establish the outlet points of the 12 catchments in which the THRB was divided (Fig. 5)……… As there are data available to the 12 gauging stations for the same period, it is possible to calibrate the model jointly for all the catchments."*

Page 9, Lines 22-24: "Among these series provided by the regionalised models, the 10 that best fitted the historical series for both temperature and precipitation were assembled on a monthly basis (...)" Authors should explain which is the control period considered, as well as the method used to test the goodness of fit and the test results. Which models were finally selected for the basin?

Thanks for the comment.

The control period considered for both models and assembles is 1971-2005. This period is more than enough to represent precipitation and temperature. About 30 years for precipitation and 17 for temperature, as recommended by the WMO (2011) (page 4-15).

The method used to test the goodness of fit and to test the results are the root mean square error ($E_{RMS}$) and the Nash-Sutcliffe efficiency criterion ($E_{NS}$), both in temperature and precipitation. In the manuscript, only the main results of $E_{NS}$ are indicated.

Among all the models (including another ensemble: Simple Average Forecast Combination), the ensemble Bias-Corrected Eigenvector Combination (EIG2) was used, since it provides a lower prediction error compared to the rest of the series. For example, the $E_{NS}$ values obtained with EIG2 for the temperatures were 0.87, while in the separate models it never exceeded 0.77. For the precipitations, the $E_{NS}$ value was 0.30, while in the models it was always lower than 0. In addition, another advantage of EIG2 method is that it allows to correct the bias produced by predictive models.

These explanations have been included in the manuscript in the section "4.1 Hydrological modelling of the Tagus Headwaters River Basin." The new third paragraph is as follow:

*"The second source was the State Meteorological Agency of Spain (AEMET), the source of the data used for the two CC scenarios of AR5 (**IPCC, 2013**). The AEMET provided the regionalised projections of 27 models (13 for RCP 4.5 and 14 for RCP 8.5), using the statistical method of analogues (**Amblar-Francés et al., 2017**). The daily temperature and precipitation series of the six thermometric and 48 precipitation stations closest to the THRB were used (**Fig. 5**). These daily data were aggregated to monthly series. Next, the reference historical data series of each model have been compared with Spain02v5 data, using as a control period the 1971-2005 interval. The 10 models that best fitted for both temperature and precipitation were assembled using the Simple Average Forecast Combination (SA), and Bias-Corrected Eigenvector Forecast Combination (EIG2) (**Hsiao and Wan, 2014**), available in the R-CRAN package*

*GeomComb (**Weiss and Roetzer, 2016**). Then, both ensembles have also been compared with Spain02v5 data using the same control period (1971-2005). Finally, the series obtained by EIG2 have been used, with a lower prediction error compared to the rest of series. For example, the $E_{NS}$ values obtained with EIG2 for the temperatures were 0.87, while in the separate models it never exceeded 0.77. For the precipitations, the $E_{NS}$ value was 0.30, while in the models it was always lower than 0. In addition, another advantage of EIG2 method is that it allows to correct the bias produced by predictive models. The period used in the simulation of CC scenarios (RCP 4.5 and 8.5) is from October 2020 to September 2090 (also seventy consecutive years)."*

This reference is not included in the manuscript:

> WMO. Guide to methodological practices. World Meteorological Organization. WMO-No. 100, 2011. Geneva.

Finally, regarding the models finally used, the 10 selected projections for the ensembles are:

ACCESS1_0, bcc_csm1_1, bcc_csm1_1_m, BNU_ESM, CNRM_CM5, inmcm4, MIROC_ESM, MIROC5, MPI_ESM_LR, MPI_ESM_MR

We consider not including this information in the manuscript so as not to extend the article further.

Page 9, Lines 25-27: "Based on the average monthly temperature data of the three climatic scenarios, the potential evapotranspiration series were estimated using the Thornthwaite method (...)". For example, it considers that PET is null when the temperature is near zero. According to the authors, "(...) while the average annual temperature is 11 â°U ¸eC, in the coldest months there are values less than zero (...)" (Page 3, Lines 29-30). As Thornthwaite method considers that potential evapotranspiration (PET) is null when the temperature is near zero, it could undervalued the PET in the Tagus basin. Authors should discuss the reasons why this approach was selected.

The process of obtaining the climatic variables has been carried out on a monthly basis, both in the historical scenario and in the future scenarios. So, we have chosen to use Thornthwaite method to calculate the PET. The problem with this method is that it tends to underestimate PET (**Gomariz-Castillo, 2015**). So, in order to correct this underestimation, the PET series generated by Thornthwaite method were corrected from a linear regression (similarly as monthly change factor (CF) used in **Guerreiro et al. (2017)**) obtained by a comparison between the estimated series and those used by the SIMPA hydrological model (**BOE, 2007**). These data of PET used by SIMPA have already been regionalized based on the Penman-Monteith equation (**Allen et al., 1998**), correcting the underestimation of the Thornthwaite method.

This explanation has also been included in the manuscript in the section "4.1 Hydrological modelling of the Tagus Headwaters River Basin". The new fourth paragraph is as follow:

*"Based on the average monthly temperature data of the three climatic scenarios, the potential evapotranspiration series were estimated using the Thornthwaite method (**Thornthwaite, 1948; Gomariz-Castillo, 2017**). As this method tends to underestimate the potential evapotranspiration, the series generated were corrected from a linear regression between the estimated series and those used by the SIMPA hydrological model (**BOE, 2007**). These series used by SIMPA have already regionalized for the Iberian Peninsula based on the Penman-Monteith method (**Allen et al., 1998**), correcting the underestimation of the Thornthwaite method."*

New references included:

Allen, R.G., Pereira, L.S., Raes, D., Smith, M.: Crop evapotranspiration- Guidelines for computing crop water requirements-FAO Irrigation and drainage paper 56, 300. Rome: FAO; p. D05109., 1998.

BOE.: Acuerdo para encomienda de gestión por el Ministerio de Agricultura, Alimentación y Medio Ambiente (Dirección General del Agua) al CEDEX, del Ministerio de Fomento, para la realización de asistencia técnica, investigación y desarrollo tecnológico en materias competencia de la Dirección General del Agua. Boletín Oficial del Estado (BOE), 287, 49436-49458. (In Spanish). https://www.boe.es/diario_boe/txt.php?id=BOE-A-2007-20623, 2007.

Page 10, Figure 5: According to the figure, only one of the considered thermopluviometric stations (3044) is located inside the basin, which according to the authors has an area of 7.000 km$^2$ (Page 3, Line 26). How could you assure that the temperature is properly characterized, considering that you only take into account 6 points (5 of them outside of the basin), for such a large area?

There is no more data of regionalized temperature scenarios in the study area. To improve its representativeness, the 6 stations have been interpolated as a previous step to their aggregation to each catchment. For this interpolation the altitude of the DEM has been taken into account as an auxiliary variable, since it has a high effect on the temperature and is also used in other works, such as Spain02v5 itself. In this process, we have checked the interpolations with the data of Spain02v5.

We include the following sentence in the first paragraph of the section "4.1 Hydrological modelling of the Tagus Headwaters River Basin".

*"… This DEM was used as an auxiliary variable in the interpolation models of the climatic variables,…".*

Have you compared the bias between the observed precipitation and temperature values and the modeled ones for the control period of the climate models?

Yes, we did it. The precipitation and temperature series have been compared with the historical reference data series and the bias was assessed. This was a measurement to select the ensemble model, the EIG2, since it evaluates and corrects the bias effects (EIG2).

These explanations have been included in the third paragraph of the section "4.1 Hydrological modelling of the Tagus Headwaters River Basin." See comment Page 9, Lines 22-24.

Page 12, Lines 3-5: "The values of the criterion coefficients calculated in the hydrological modelling show that the model employed reproduced properly the surface flows in the THRB: high values of ENS and R2, together with low relative errors (ERMS) and volume errors (PBIAS)". According to Table 2, calibration values are in the range of 0,51-0,87 for the NSE. Nevertheless, NSE values for the validation period oscillate between -0,36 and 0,57. Moreover, 10 out of 12 sub-basins show NSE values below 0,50 during the validation period. Regarding these results, it is not possible to state that the hydrological model is properly reproducing the surface flows during the validation period. Authors should consider model re-calibration or even the use of another hydrological model.

Thanks for this reflection. The referee #2 also refers to this question indicating that it could be explained by the uncertainty of the observed flows data. Although the data series used come from official sources (yearbook of Official Gauging Station Network of Spain), and they have been purified before their publication, it could be possible some errors in the measurements. If there would be errors in the measurements, they would be more probable in the older data

since the gauging stations could be not properly calibrated. In this case, errors can have more influence on the results in the gauging stations of the upper catchments since they have lower flows. It could also explain the results of the simulation of the Buendía reservoirs inflows by Lobanova et al. (2016). They obtained a worse goodness of fit in the calibration (NSE = 0.39 for the period 1987-1993) than the validation period (NSE = 0.76 for the period 1994-1999). In addition, these results are in line with those of this work.

Besides this source of uncertainty (errors in the measurements), we can add the fact of using the validation period just after the warming up period (the last being the calibration period). Thereby, it could cause that, in some basin, the warming up period can extend to a part of the validation period. This would entail that the calibration process used the warming up period and a part of the validation to adjust the initial parameters, obtaining worse adjustments in the validation of the model. The union of both sources of uncertainty may be the reasons why the validation yields low values in the goodness of fit. However, as the ultimate goal is to use a calibrated model for the last few years, so the result obtained in the performance is considered as good.

We include a reflection about this issue in the first paragraph of the section "5.1. Climate change effects on the hydrology of the Tagus Headwaters River Basin".

*"The values of the criterion coefficients calculated in the hydrological modelling show that the model employed reproduced properly the surface flows in the THRB in the calibration period: high values of $E_{NS}$ and $R^2$, together with low relative errors ($E_{RMS}$) and volume errors ($P_{BIAS}$). However, in the validation period, there are some low values for the goodness of fit coefficients calculated, indicating that the results of these catchments have greater uncertainty. These results can be explained by the fact that of using the validation period just after the warming up. Thereby, it could cause that, in some catchments, the warming up can extend to a part of the validation period. This would entail that the calibration process used a part of the validation to adjust the initial parameters, obtaining worse adjustments in the validation of the model. But, it is important to highlight that the parameters used in the simulations are adjusted with the more recent data, providing a good performance of the surface flows in the THRB. In addition, the best performance in calibration corresponded to the outlets of the catchments of Entrepeñas and Buendía, which were the flows used as the input in the subsequent simulation of the water resources exploitation system. In fact, both catchments had $N_{SE}$ values around 0.80 and low $P_{BIAS}$ (**Table 2**)."*

Page 12, Lines 8-11: "The simulation of the historical climate series of 1940-2010 with the calibrated model provided an average annual resource of 954.6âˇA ´c106m3yr☐1 (Q). These series was temporally translated to the 2020-2090 time period in order to reproduce a future climate scenario without climate change (No CC). The simulations for climate change scenarios RCP 4.5 and RCP 8.5 with the same calibrated model indicated that the THRB could suffer a considerable loss of its natural water resources". The meaning of "temporally translated" is not clear. Are reduction/increase coefficients obtained from the RCPs applied to the precipitation/temperature 1940-2010 time series to simulate "future conditions"?

Thanks for the question. No coefficients have been applied to the precipitation/temperature 1940-2010 time series. We assume that if the climate does not change, the future climate will be the same as the past. Then, we simulate the hydrological semi-distributed model calibrated

with the observed flows from 1995 to 2009 using the precipitation and temperature series of 1940-2010.

We have clarified this question better in sections "3. Methodology" and "4.1 Hydrological modelling of the Tagus Headwaters River Basin".

3. Methodology:

*"… In the first one the historical climate series were used (No CC) without climatic correction coefficients. …"*

4.1 Hydrological modelling of the Tagus Headwaters River Basin (second paragraph):

*"… For the No CC scenario, these series of data were extended from October 1940 to September 2010 (seventy consecutive years), and they were used in the simulation without climatic correction coefficients. …"*

Page 12, Lines 12-15: "The RCP 4.5 scenario forecasted a value of 575.6ā˘A´c106m3yr☐1, representing a decrease of 39.7 %, while the RCP 8.5 scenario predicted a 46.6 % decline in resources to 508.9ā˘A´c106m3yr☐1, on average (Fig. 7). This is due to a combination of a reduction in precipitation (15% and 20% for each scenario, respectively) and an increase in potential evapotranspiration resulting from an increase in temperature of 2.2_C and 3.4_C, respectively, for the CC scenarios RCP 4.5 and 8.5". These results should be contrasted and discussed in relation to previous studies of climate change impacts under RCP scenarios in the basin (CEDEX, 2017; Guerreiro et al., 2017; Lobanova et al., 2016; Lobanova et al., 2018).

Thanks for the recommendation. We have contrasted the results with those in the indicated references. We have extended the first paragraph about this comparison in the section "6. Discussion":

*"The hydrological modelling carried out with the AR5 CC scenarios predict a sharp decrease in the water resources of the THRB. These results are in line with the previous simulations made on the same scale (Lobanova et al., 2016), and on a larger scale as well (Guerreiro et al., 2017; CEDEX, 2011b; Lobanova et al., 2018). The hydrological modelling indicate that the increase in temperature would generate a decline in snowfall, which leads to a reduction in the snow cover period of two months. The main snowfall would have a delay of one month and the snowmelt would start a month earlier. The decreases provided by the simulations are similar to those published by CEDEX (2017) for the same mountainous area, in which a decrease of up to 70% is predicted for the interval 2070-2100 in the RCP 4.5 scenario, and greater than 90% for the RCP 8.5 scenario (same interval). In addition, CEDEX (2017) also predicts a decrease in the two-month snow cover period. Regarding the aquifer recharge, the hydrological modelling made indicate a decline greater than 50%. Although these values are slightly higher than those published by CEDEX (2017) for the whole Tajo River Basin, they are in line with the previsions in the THRB, since the majority of the projections used for these two scenarios (RCP 4.5 and RCP 8.5) indicate diminishments around 50% in aquifer recharge of this area. These alterations of the hydrological processes will probably generate a change in the fluvial regime. In relative terms, the months of autumn (October, November, December) would suffer the major decrease, this change being consistent with the results of Lobanova et al. (2018). At the annual level, these alterations would result in greater variability of the surface flows of the basin due to increases in the relative difference between the maximum and minimum flows of spring and*

*summer, respectively. In short, these changes are going to diminish the natural capacity of the THRB to regulate its own water resources. However, the results of the THWRES simulations indicate that the EBR will not reach their maximum volume in any situation, and no new infrastructure will be necessary to overcome this loss of regulation."*

This reference has also been included:

CEDEX: Evaluación del Impacto del Cambio Climático en los Recursos Hídricos y Sequías en España. Informe Final. Centro de Estudios Hidrográficos. Ministerio de Agricultura, Alimentación y Medio Ambiente. Madrid. Spain. (In Spanish), http://www.cedex.es/CEDEX/LANG_CASTELLANO/ORGANISMO/CENTYLAB/CEH/Documentos_Descargas/EvaluacionimpactoCCsequiasEspana2017.htm, 2017.

Page 18, Lines 1-5: "The whole irrigated area in the SRB is divided into seven irrigation zones (IZs); for each of them the water demand curve is estimated from a linear programming model that optimises the Gross Value Added (GVA) by the optimal cultivation plans according to the water supply. This crop programming includes the irrigation situations of woody crop maintenance, the change from irrigated to rainfed crops and the abandonment of irrigation polygons, as well as the impact on employment". Demand curves are mentioned for the first time in the Discussion section. If these demand curves are obtained by the authors, further insight should be provided in the Methodology section, and plots included in the Results section. In any other case, sources should be properly referenced.

Thanks for the suggestion. In the work, the section dedicated to the study of the socioeconomic impact has been enlarged. At the end of the section "1. Introduction", we have included the socioeconomic study as a complementary objective to the work.

[revised manuscript text omitted]

As the socioeconomic study is considered a complementary objective, their results are presented in the section "6. Discussion".

*"The socioeconomic impact of the decreases in the TSA supply calculated in the present work can be approximated by the methodology presented at the section 3.3. In the first place, the global decrease in TSA flows to the SRB has been distributed among seven IZs, according to the results obtained by Martinez-Paz et al. (2016) for the repercussion of supply deficits in them, conditioned by both their water demand and the typology of the exploitation system of the SRB. Once these deficits had been determined, they were valued using the linear programming presented, calculating GVA and agricultural employment for each IZ according to water allocations available*

in each scenario. The results are presented in Table 4, which shows the differences in relation to scenario with no climate change (No CC).

Table 4. Differences of RCP 4.5 and RCP 8.5 scenarios respect to No CC (these values represent decreases with respect to No CC scenario).

| IZ | RCP 4.5 | | | RCP 8.5 | | |
|---|---|---|---|---|---|---|
| | TSA to SRB ($10^6 \cdot m^3$/year) | GVA ($10^6$€/year) | Employment ( full-time jobs/ year) | TSA to SRB ($10^6 \cdot m^3$/year) | GVA ($10^6$€/year) | Employment ( full-time jobs/ year) |
| 1 | 54.60 | 79.12 | 1135 | 60.14 | 88.61 | 1251 |
| 2 | 61.60 | 58.33 | 455 | 67.85 | 64.26 | 502 |
| 3 | 22.58 | 34.04 | 651 | 24.87 | 38.65 | 717 |
| 4 | 44.99 | 68.49 | 1484 | 49.55 | 77.26 | 1635 |
| 5 | 93.12 | 122.25 | 2582 | 102.57 | 135.06 | 2845 |
| 6 | 9.41 | 11.51 | 378 | 10.37 | 13.82 | 416 |
| 7 | 8.21 | 5.94 | 3 | 9.04 | 6.79 | 4 |
| SRB | 294.5 | 379.67 | 6690 | 324.4 | 424.46 | 7369 |

By calculating the ratio between the GVA and TSA the marginal value of water is obtained: 1.29 €/m³ for the RCP 4.5 scenario and 1.31 €/m³ for the RCP 8.5 scenario, for the whole of the SRB irrigation. The marginal value marks the upper limit of the marginal productivity of water, and therefore it is the maximum payment capacity of the sector for water in each scenario. The obtained figures are in range with those presented in other studies for the area (Martínez-Paz et al, 2016; Albaladejo et al, 2018).

Thus, the decrease in the TSA supply for the RCP 4.5 scenario would mean a direct loss of 380 million € per year (at the prices of 2017), while for RCP 8.5 it would amount to 425 million €/year. Given that the GVA of irrigation in the SRB is around 1870 million € (Martinez-Paz and Pellicer-Martinez, 2018), these figures represent, respectively, direct losses of 20% and 23% in terms of GVA. In addition, the decrease in the volumes transferred in the CC scenarios means that the irrigated area is occupied by more-labour-extensive crops, causing the loss of around 7000 direct full-time agrarian jobs per year in the sector in each situation. To all these direct effects we should add the drag effects of the primary sector on other economic sectors that depend directly on the use of irrigation in the area (food industry, marketing, transport, inputs, etc.) and make up the well-known agro-industrial cluster of the Region of Murcia (Colino et al., 2014). In this sense, some authors propose a multiplier of no less than 3 to calculate the total economic impact of agricultural production on other related activities. To this economic impact it would be possible to add the environmental one, as it would be, for example, the reduction of the flows circulating in the Segura River, since it is part of the distribution network of the TSA volumes, which would also have effects on the ecosystems associated with it (Perni and Martinez-Paz, 2017)."

Finally, as we have changed the operating rule, there have been small variations in the transferred flows in each scenario, so there are also little variations in the socioeconomic impacts respect to the previous version.

We have included these new references:

Albaladejo-García, J.A., Martínez-Paz, J.M., Colino, J.: Financial evaluation of the feasibility of using desalinated water in the greenhouse agriculture of Campo de Níjar (Almería, Spain) (In Spanish) ITEA-Inf. Tec. Econ. Ag., (In press), 2018

Griffin, R.C. (2006). Water resource economics: the analysis of scarcity, policies and projects. Massachusetts Institute of Technology. Cambridge.

OTHER COMMENTS

- There is a misspelling in one of the authors' names: it should be Francisco instead of Fancisco.

Thanks for the appreciation. We have corrected the errata.

- Figure 6. The text does not match the figure (it represents the DSS diagram instead of the distribution of climate stations with AR5 information and the grid obtained from Spain 02v5.

Thanks for the observation. It has been a misprint. We have corrected the title of this Figure that coincided with Figure 5.

The correct caption is:

*"Fig. 6. Scheme of the water resources exploitation system of the Tagus Headwaters, as far as Aranjuez"*

[Figure]

- Acknowledgments: As Spain02 v5 datasets have been used, authors should include the following text: "The authors thank AEMET and UC for the data provided for this work (Spain02 v5 dataset, available at http://www.meteo.unican.es/datasets/spain02)"

Thanks for the recommendation. We have included this phrase in the acknowledgments.

- Page 20. Lines 13-18. There is an updated work from the same institution, which uses the same climate models considered in the present work: CEDEX, 2017. Evaluación del impacto del cambio climático en los recursos hídricos y sequías en España. Informe final. Ministerio de Agricultura, Alimentación y Medio Ambiente.

Thanks for the recommendation. We have included this reference.

- Additional suggested references:

Anastasia Lobanova, Stefan Liersch, Joao Pedro Nunes, Iulii Didovets, Judith Stagl, Shaochun Huang, Hagen Koch, María del Rocío Rivas López, Cathrine Fox Maule, Fred Hattermann, Valentina Krysanova, Hydrological impacts of moderate and high-end climate change across European river basins, Journal of Hydrology: Regional Studies, Volume 18, 2018, Pages 15-30, ISSN 2214-5818, https://doi.org/10.1016/j.ejrh.2018.05.003.

Enze Zhang, Xin'an Yin, Zhihao Xu, Zhifeng Yang. Bottom-up quantification of interbasin water transfer vulnerability to climate change, Ecological Indicators, Volume 92, 2018, Pages 195-206, ISSN 1470-160X, https://doi.org/10.1016/j.ecolind.2017.04.019.

Liz Clarke, Mark Rousevell, Miriam Dunn et al. Climate Change Impacts, Adaptation and Vulnerability Model Applications in Three Regional to Local Scale Case Studies in Europe. IMPRESSIONS Deliverable D3C.2. July 2017. Available online:

Selma B. Guerreiro, Stephen Birkinshaw, Chris Kilsby, Hayley J. Fowler, Elizabeth Lewis. Dry getting drier – The future of transnational river basins in Iberia, Journal of Hydrology: Regional Studies, Volume 12, 2017, Pages 238-252, ISSN 2214-5818, https://doi.org/10.1016/j.ejrh.2017.05.009.

Thanks for the recommendation. We have included three of these references, and the third one have been updated, since it has not the number of the pages nor the year. Thanks.